# Dynamic nucleolar phase separation influenced by non-canonical function of LIN28A instructs pluripotent stem cell fate decisions

Tianyu Tan[1,2], Bo Gao[3], Hua Yu[1], Hongru Pan[1], Zhen Sun[1], Anhua Lei[1], Li Zhang[1], Hengxing Lu[2], Hao Wu[4], George Q. Daley [5], Yu Feng [3] ✉ & Jin Zhang [1,2,6,7] ✉

LIN28A is important in somatic reprogramming and pluripotency regulation. Although previous studies addressed that LIN28A can repress *let-7* microRNA maturation in the cytoplasm, few focused on its role within the nucleus. Here, we show that the nucleolus-localized LIN28A protein undergoes liquid-liquid phase separation (LLPS) in mouse embryonic stem cells (mESCs) and in vitro. The RNA binding domains (RBD) and intrinsically disordered regions (IDR) of LIN28A contribute to LIN28A and the other nucleolar proteins' phase-separated condensate establishment. S120A, S200A and R192G mutations in the IDR result in subcellular mislocalization of LIN28A and abnormal nucleolar phase separation. Moreover, we find that the naive-to-primed pluripotency state conversion and the reprogramming are associated with dynamic nucleolar remodeling, which depends on LIN28A's phase separation capacity, because the LIN28A IDR point mutations abolish its role in regulating nucleolus and in these cell fate decision processes, and an exogenous IDR rescues it. These findings shed light on the nucleolar function in pluripotent stem cell states and on a non-canonical RNA-independent role of LIN28A in phase separation and cell fate decisions.

Somatic cells can be induced to the pluripotent state[1], and different pluripotent states - the naive and primed states can be interconverted[2]. Numerous studies have been carried out to characterize the genetic, epigenetic and other cellular events associated with reprogramming and naive-to-primed state conversion[3–5], and LIN28A was found to be an important factor mediating both processes[6,7]. *Lin28* was discovered as a heterochronic gene in nematode *Caenorhabditis elegans*, involved in regulating the developmental timing of *c. elegans*[8]. It was also found to facilitate reprogramming with OCT4, SOX2, and NANOG[6], and required for the primed state conversion in mouse embryonic stem cells (ESCs)[5,7]. More studies showed that LIN28A affected cell proliferation and regulated glucose metabolism by binding and inhibiting biogenesis of the tumor suppressor *let-7* family microRNA[9–12]. Recently, it was shown that LIN28A was present in the mammalian

[1]Center for Stem Cell and Regenerative Medicine, Department of Basic Medical Sciences, and Bone Marrow Transplantation Center of the First Affiliated Hospital, Zhejiang University School of Medicine, Hangzhou 310003, China. [2]Liangzhu Laboratory, Zhejiang University, Hangzhou 310000, China. [3]Department of Biophysics, and Department of Infectious Disease of Sir Run Run Shaw Hospital, Zhejiang University School of Medicine, Hangzhou 310058, China. [4]Program in Cellular and Molecular Medicine, Boston Children's Hospital, Department of Biological Chemistry and Molecular Pharmacology, Harvard Medical School, Boston, MA, USA. [5]Stem Cell Transplantation Program, Division of Pediatric Hematology Oncology, Boston Children's Hospital, Department of Biological Chemistry and Molecular Pharmacology, Harvard Medical School, Boston, MA, USA. [6]Institute of Hematology, Zhejiang University, Hangzhou 310058, China. [7]Center of Gene/Cell Engineering and Genome Medicine, Hangzhou 310058, China. ✉e-mail: yufengjay@zju.edu.cn; zhgene@zju.edu.cn

nucleolus during early embryonic development[13]. Despite the long history since the initial discovery of the nucleolus[14], its functions and mechanisms in development and stem cell fate decision are yet to be completely uncovered. It was found that the nucleolus had liquid-phase separation[15]. A typical nucleolus is a compartmentalized structure in which many proteins reside in. These nucleolar proteins form complexes and contribute to the FC (marker: RPA194), DFC(marker FBL, NCL) and GC (marker: NPM) nucleolar liquid phase stratification[14,16–18]. In the naive pluripotent state ESCs, LIN28A was shown to be mainly distributed in the GC and FC layers of the nucleolus, and got redistributed to be preferably in the cytosol upon conversion from the naive to the primed state[19]. In the nucleolus of naive ESCs, LIN28A maintains nucleolar integrity and helps repress the 2CLC-associated genes[19]. In the naive-to-primed ESC conversion, LIN28A regulates the metabolic program associated with the naive and primed state[5,7]. It is unknown how nucleolus and its phase separation capacity remodel during the naive-to-primed conversion and the reprogramming and whether LIN28A plays a role in these nucleolar remodeling processes.

LIN28A proteins are conserved in many species. It contains a cold shock domain (CSD) and two CCHC-type zinc knuckle domains (ZKD). Both domains can bind with RNA. LIN28A binding and repression of let-7 relies on both CSD and CCHC zinc fingers. Mutations in these RNA binding regions abrogated the let-7 inhibition function of LIN28A[20,21]. Mutations in the C-terminal intrinsic disordered region (IDR) have also been reported. For instance, phospho-null mutation LIN28A (S200A) decreased protein stability which led to reduced reprogramming efficiency[22]. Another mutation R192G in the IDR has been reported to be associated with Parkinson disease pathogenesis[23]. Whether the IDR regions and the mutations within these regions influence LIN28A phase separation and subsequently regulate nucleolar integrity and functions, and if so, whether LIN28A can regulate stem cell fate decision through its nucleolar phase separation property are unknown.

In this study, we observed that the linker and C-terminal regions of LIN28A, which include S120, R192 and S200, were predicted as the intrinsically disordered regions (IDR). They promoted the establishment of the LIN28A protein phase-separated condensate. Moreover, LIN28A interacted with nucleolar proteins FBL and NCL to facilitate the nucleolar phase separation, and its RBD or IDR-truncated mutations, as well as the three IDR point mutations, led to nucleolar defects and a block of primed pluripotency conversion in mouse embryonic stem cells and reduced reprogramming efficiency from somatic cells. The nucleolus liquid-liquid phase separation (LLPS) model of LIN28A provided insights into how the non-canonical function of the RNA-binding protein and pluripotent factor can contribute to nucleolar integrity and consequently determine the pluripotent stem cell fate decision.

## Results

### Nucleolar LIN28A protein undergoes phase separation, and is temperature-sensitive

In order to study LIN28A in living cells, we generated the eGFP-LIN28A knock-in ESC line and eGFP-LIN28A-overexpressing ESC line (Fig. 1a and Supplementary Fig. 1a). LIN28A immunostaining in the wild-type mESCs, or live imaging of LIN28A in the eGFP-LIN28A-overexpressing and eGFP knock-in ESCs all showed that a great portion of fluorescence-labeled eGFP-LIN28A protein is clearly condensed in the nucleolus (Fig. 1b). We tested the fluidity of LIN28A protein, fluorescence recovery after photobleaching (FRAP) experiments indicated that overexpressed LIN28A exhibited similar fluidity with the knock-in LIN28A both in the nucleolus and cytoplasm (Supplementary Fig. 1b). Overexpressed LIN28A and eGFP knock-in LIN28A had the similar localization and behavior.

Moreover, immunostaining by Stimulated Emission Depletion Microscopy(STED) showed co-localization of LIN28A with DFC(FBL)

and GC(NPM) in the nucleolus in E14 mESCs cultured in LIF/serum. LIN28A covered a larger region around FBL with empty holes in the middle, consistent with previous study that LIN28A was distributed mainly in the GC and DFC regions (Fig. 1c).

Mammalian nucleolus was a multiphase liquid condensate[14]. The aliphatic alcohol 1,6-hexanediol(HEX) interferes with weak hydrophobic interactions and is often used to dissolve protein phase separated condensates in cells[24]. LIN28A is present both in the nucleolus and cytoplasm. Thus we used HEX to study in which compartment can LIN28A form phase separated condensates. Endogenous eGFP-LIN28A knock-in ESCs treated with 1% HEX for 10 minutes showed diffusion of the condensates in the nucleolus, and the nucleolar LIN28A condensate was more sensitive to the HEX treatment (Fig. 1d). The statistical analysis also quantitatively showed a reduction in LIN28A condensates intensity in the nucleolus after HEX treatment (Fig. 1e), and an increase in the dispersed area and irregularity of LIN28A in the nucleolus after HEX treatment (Fig. 1f, g). HEX treatment did not affect cytoplasmic LIN28A distribution, suggesting that cytoplasmic LIN28A was more diffused and did not have typical phase-separated condensate behavior.

Next, we determined whether the LIN28A phase-separated condensate was affected by temperature. When cells were exposed to cold shock (25 °C) for 30 min, LIN28A tended to be reduced in the nucleus (Supplementary Fig. 1e; middle). When cells were exposed to heat shock (42 °C) for 15 minutes, LIN28A tended to become more compact compared with that at 37 °C (Supplementary Fig. 1e; right). We investigated the difference in fluidity of the LIN28A phase-separated condensate at 37 °C, 25 °C and 42 °C. Fluorescence recovery after photobleaching (FRAP) analysis of eGFP-LIN28A was performed at the indicated temperatures. After bleaching for 300 seconds, fluorescence signals of LIN28A in the nucleus had 20–30% of recovery in all the three conditions, with no significant difference in the degree of LIN28A recovery (Supplementary Fig. 1f; up,1 g). In contrast, the fluidity of LIN28A in the cytoplasm had above 40% recovery and the 42 °C condition had over 60% recovery (Supplementary Fig. 1f; down, 1 g). Statistical analysis showed that cold shock decreased LIN28A loci numbers in the nucleus and the nucleus: cytoplasm intensity ratio, whereas heat shock increased both (Supplementary Fig. 1h, i). We also observed more dispersed areas of LIN28A in cold-shocked nuclei and more compact areas of LIN28A in heat-shocked nuclei (Supplementary Fig. 1j). Importantly, the different temperatures did not affect the whole expression level of LIN28A (Supplementary Fig. 3f). These results suggested that cold shock promoted LIN28A outflow from the nucleus and the condensate became more diffused, while heat shock promoted its inflow into the nucleus and the condensate became more compact.

Together, these results demonstrated that LIN28A protein in the two subcellular compartment assumes different states-the cytosolic LIN28A has more liquid-like distribution, and the nucleolar LIN28A forms more solid-like phase-separated condensates[25,26].

### LIN28A undergoes RNA-dependent phase separation in vitro

Given that the RNA-binding protein LIN28A formed condensates in living cells, we examined whether it also underwent LLPS in vitro. LIN28A did not form LLPS with 150 mM high concentration (physiological salt conditions) of NaCl (Supplementary Fig. 1c, d), and its phase separation was observed at a concentration of 50 mM NaCl in the presence of total RNA (Fig. 1h, i and Supplementary Fig. 1c, d). Next, we sought to enrich rRNA which is in the nucleolus from mouse ESCs and mixed LIN28A protein (250 μM) with rRNA (50 ng/μl) and observed that LIN28A and rRNA mixture can form visualized phase separated droplets, and the droplets were more compact (Fig. 1j). Overall, these results indicated that LIN28A underwent RNA-dependent phase separation in vitro. Notably, LIN28A was more susceptible to form droplets in the presence of rRNA, suggesting that LIN28A phase separation might play an important role in the nucleolus of pluripotent stem cells.

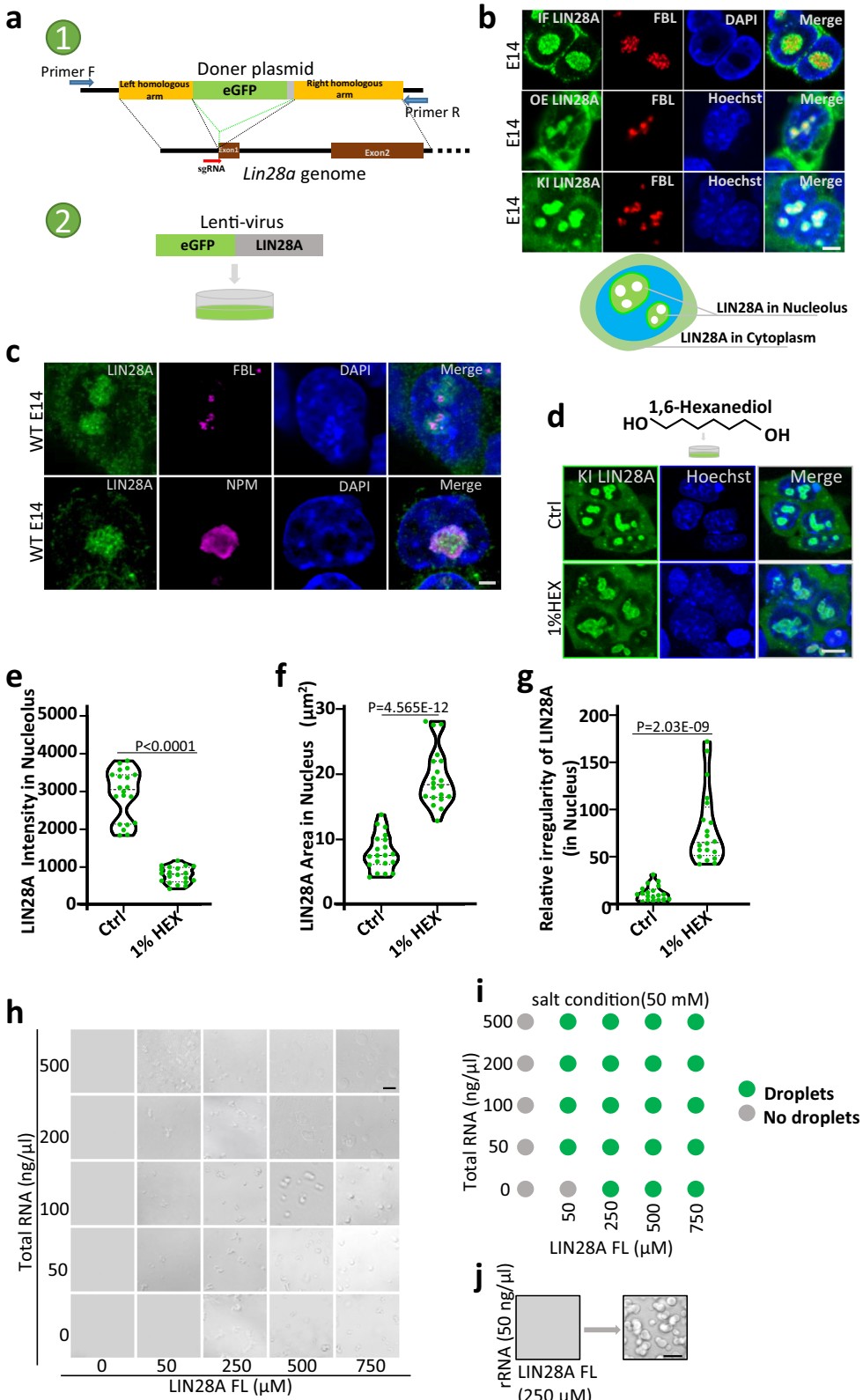

## rRNA is essential to maintain the localization and fluidity of LIN28A in the nucleolus

As rRNA facilitated LIN28A phase separation in vitro illustrated above, we wondered whether rRNA played a role in maintaining the localization and phase-separated state of LIN28A in the nucleus in cells. Thus, three ways of inhibiting rRNA were employed: (1) RNA polymerase I

(Pol I) inhibitor CX-5461 small molecule treatment, (2) Pol I degradation system with the degron technology[27] (Supplementary Fig. 3c, d), and (3) a snoRNA knockout line. Cells treated with $2\,\mu M$ CX-5461 for 12 h had significantly reduced rRNA levels, which led to reduced or even absence of LIN28A in the nucleoli (Fig. 2a). When Pol I was degraded by an auxin-inducible degron system[27], LIN28A was no

**Fig. 1 | Nucleolar LIN28A protein is sensitive to the HEX treatment in vivo and undergoes RNA-dependent phase separation in vitro. a** schematic diagram showing the nucleolus of E14 mouse ESC line with eGFP knock-in(KI) or over-expression(OE) to create the LIN28A fusion proteins. **b** Co-localization of LIN28A and FBL in the nucleolus in immunostaining E14 mESCs, eGFP knock-in E14 mES cells and eGFP-LIN28A over-expression E14 mESCs. Scale bar, 5 μm. **c** Immunostaining showing co-localization of LIN28A with DFC(FBL), and GC(NPM) by Stimulated Emission Depletion Microscopy(FBL and NPM applied STED pattern). Scale bar, 2 μm. **d** Confocal microscopy Airyscan imaging of the living Knock-in eGFP-LIN28A ESCs with 1%HEX treatment. Scale bar, 10 μm. **e** Statistical analysis of nucleolar LIN28A fluorescence intensity with and without the 1%HEX treatment; $n = 20$ cells, two-tailed unpaired student's $t$-test. **f** Statistical analysis of LIN28A loci

area in the nucleolus with and without the 1%HEX treatment. $n = 20$ cells, two-tailed unpaired student's $t$-test. **g** Statistical analysis of LIN28A relative irregularity in nucleolus with and without the 1%HEX treatment; $n = 20$ cells, two-tailed unpaired student's $t$-test. **h** In vitro phase separation assay was performed with various concentrations of LIN28A protein and RNA in reaction buffer containing 50 mM HEPES pH 7.5, 50 mM NaCl, 1 mM DTT, and 10% PEG-8000 (Sigma). RNA purified from mouse ESCs. Scale bar, 10 μm. **i** Summary of LLPS of LIN28A under indicated conditions, in the presence of 50 mM NaCl in vitro. **j** LIN28A protein and rRNA form liquid droplets in vitro. Scale bar, 10 μm. For (**b, c, h, j**) three times biologically independent experiments were performed. Source data are provided as a Source Data file.

longer located in the nucleoli (Fig. 2a). Lastly, as snoRNAs are important for rRNA modification and biogenesis, in the snoRNA knockout ESCs, LIN28A tended to be reduced in the nucleolus (Fig. 2a). The statistical analysis also quantitatively showed a reduction in LIN28A condensates loci number and intensity in the nucleoli in the three conditions of rRNA inhibition (Fig. 2d, e), and an increase in the dispersed area of LIN28A in the nucleus of snoRNA KO cells (Fig. 2f). To determine the difference in fluidity of the phase-separated LIN28A upon rRNA inhibition, FRAP analysis of overexpressed eGFP-LIN28A was performed at the above experimental conditions. They all showed reduced degree of recovery of LIN28A in the cytoplasm, and the fluidity of the remaining LIN28A in the nucleolus of snoRNA KO cells was also lower, suggesting the rRNA and its interaction with LIN28A might be affected (Fig. 2b, c). Importantly, all three ways of rRNA inhibition did not affect the whole expression level of LIN28A (Supplementary Fig. 3e). Together, these results demonstrated that rRNA is essential in promoting LIN28A phase separation and maintaining its localization in the nucleoli.

### rRNA is essential to maintain the phase separation of nucleolar proteins associated with LIN28A in mouse ESCs

It is well-established that the nucleolus consists of three phase separated subcompartments: the fibrillar center (FC), the dense fibrillar component (DFC), and the granular component (GC)[14,16,17,28,29]. Each fibrillar center (FC) contains transcriptionally active ribosomal DNAs (rDNAs) and is surrounded by mini liquid droplets of pre-rRNA processing factors that are further assembled into the DFC[14]. Our previous study reported that LIN28A overlapped with the inner DFC (marked by FBL or NCL), where rRNA is modified[19]. To further understand the effects on LIN28A-associated nucleolar protein phase separation in ESCs with rRNA biogenesis defect, we constructed FBL-mCherry /eGFP-LIN28A fusion protein expressing stable cell lines for real-time monitoring (Fig. 2g). We observed that CX-5461 treatment led to an obvious nucleolar disruption with an appearance of 'grotesque nucleoli'. LIN28A greatly reduced in the nucleolus, and FBL became a sharp dot (Fig. 2h). When we used the Pol I degradation system to abolish rRNA synthesis, we found LIN28A escaped from the nucleolus, and FBL was distributed more sporadically. This reflected a severely disrupted nucleolar phase separation with an appearance of 'fragmented nucleoli' (Fig. 2h). Given that rRNA modification mediated by snoRNA took place at the DFC, we further examined FBL/LIN28A protein nucleolar phase separation in the snoRNA KO cell line. In this case, although LIN28A did not completely disappear from the nucleolar, the nucleolar phase separation showed abnormal 'diffused nucleoli' (Fig. 2h). Statistical analysis of the numbers of the representative morphology of FBL in the above cells quantitatively revealed the impairment of the nucleolar integrity (Fig. 2i). Overall, the nucleolar LIN28A condensate was sensitive to the rRNA inhibition, and it could be used as a marker of nucleolar integrity.

Next, we investigated another nucleolar DFC marker protein NCL. NCL-eGFP/mCherry-LIN28A fusion protein expression stable cell lines were constructed for real-time monitoring (Supplementary Fig. 2a).

Remarkably, the 'ring' structure-like condensate of NCL exhibited more severe disruption than that of FBL. CX-5461 treatment caused NCL to have an appearance of 'nucleolar caps'[30] (Supplementary Fig. 2b). Upon RNA Pol I degradation, NCL almost completely left the nucleoli and became scattered in the cytoplasm (Supplementary Fig. 2b). In snoRNA KO cells, NCL showed an abnormal shape, similar to 'small beads' (Supplementary Fig. 2b). Statistical analysis of the numbers of the representative morphology of NCL, and inner diameter of NCL ring in the above cells quantitatively revealed the impairment of nucleolar integrity (Supplementary Fig. 2c, d).

Together, these results demonstrated that rRNA is essential to maintain nucleolar protein phase separation, and LIN28A, like the well-known nucleolar marker proteins FBL and NCLs, can be used as a marker to indicate the integrity of nucleolar phase separation.

### Both RBDs and IDRs of LIN28A are essential for nucleolar protein LLPS

LIN28A has five major domains: an N-terminal (amino acids 1–38), predicted to be an intrinsically disordered region which we designated as IDR1; a cold shock domain (CSD) and a cluster of two CCHC-type zinc finger motifs(ZFD), which had been shown to bind RNA; a flexible linker (amino acids 113–136) between CSD and ZFD, which we designated as IDR2, and; a C-terminal (amino acids 177–209) predicted to be a disordered region, which we designated as IDR3 (Fig. 3a).

To determine the contribution of an individual domain of LIN28A to nucleolar LLPS, we generated constructs in which the N-terminal, C-terminal, both N/C- terminal, the CSD (amino acids 39–112) and the ZFD (amino acids 137–176) were individually deleted. First, we introduced the FBL-mCherry or NCL-eGFP fusion protein in a LIN28A knockout ES single clone line generated by the CRISPR/CAS9 technology (Supplementary Fig. 3a, b), then the truncated LIN28A variants fused with eGFP or mCherry were introduced to the lines (Fig. 3b and Supplementary Fig. 4a).

LIN28A knockout resulted in nucleolar disruption with a disseminated FBL and NCL appearance. The defects could be rescued by overexpressing full-length LIN28A, as FBL and NCL resumed their dense 'ring' structure. The deletion of IDR1or IDR3 or both led to nucleolar disruption, indicated by an appearance of abnormal diffused FBL and NCL (Fig. 3c and Supplementary Fig. 4b). Deletion of CSD or ZFD seriously disrupted the normal morphology of LIN28A and its co-localization with FBL or NCL in the nucleolus (Fig. 3c and Supplementary Fig. 4b). In brief, both FBL and NCL lost their 'ring' structure when either IDR or RBD of LIN28A was truncated. Furthermore, we examined the fluidity of FBL and NCL in the truncated LIN28A cell lines and found that both FBL and NCL exhibited lower fluidity in the truncated LIN28A variant cells (Fig. 3e, f, and Supplementary Fig. 4c). Statistical analysis of the numbers of the representative morphology of FBL and NCL in the truncated LIN28A variant cells quantitatively revealed impaired nucleolar integrity (Fig. 3d and Supplementary Fig. 4d).

Next, in the in vitro assay, we purified truncated LIN28A variant proteins (Fig. 3g), and found that deletion of any of the above IDR or

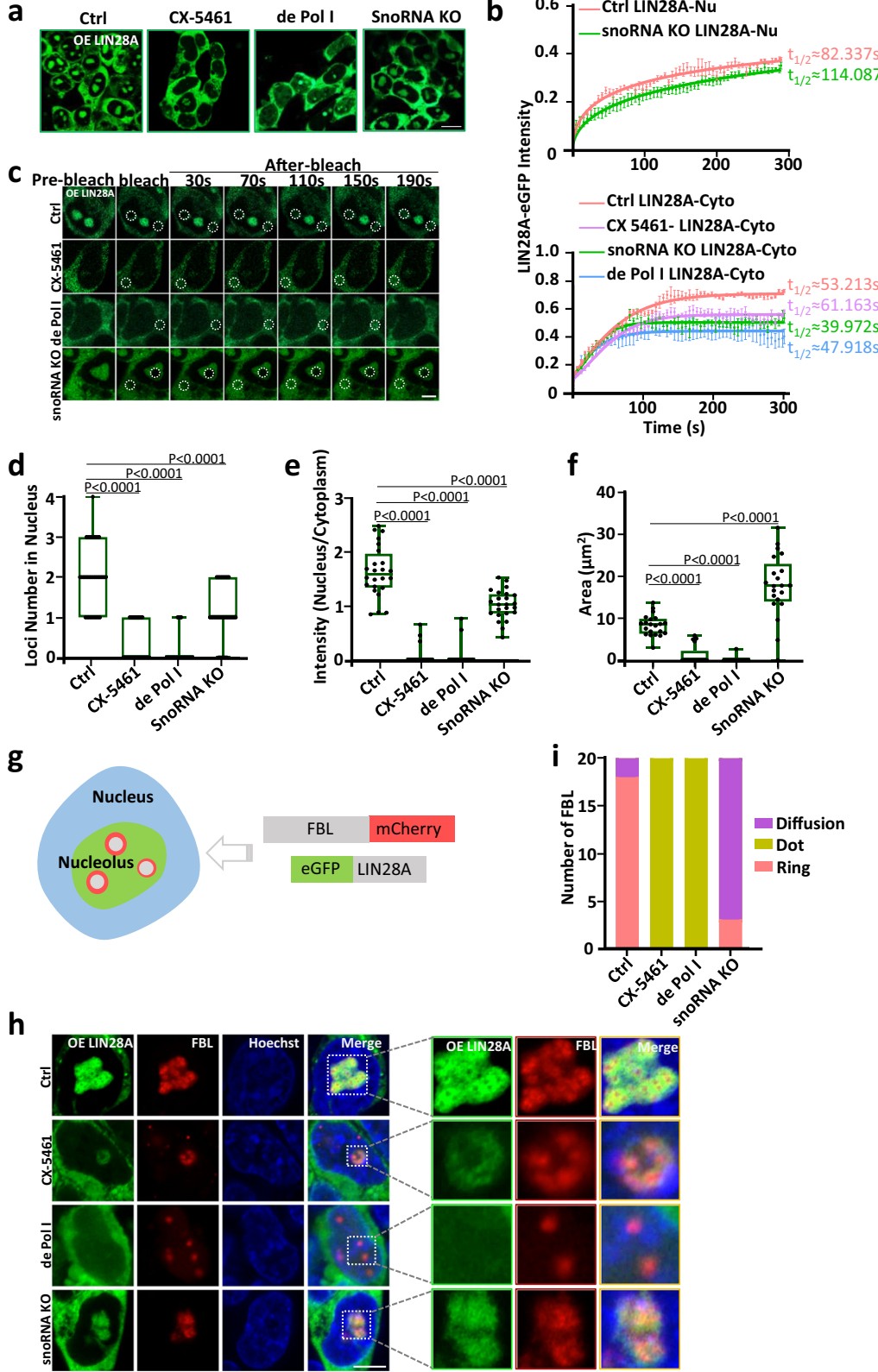

RBD domain abolished the LLPS of LIN28A in the presence of RNA (Fig. 3h, i).

Together, these data demonstrated that both the LIN28A IDR and RBD domains were important for forming the phase-separated structure of LIN28A itself and for promoting the LIN28A-mediated nucleolar protein phase separation.

## Key amino acids at the IDR region are essential for LIN28A and nucleolar phase separation

Next, we further narrowed down to the key amino acids essential for conferring the phase separation property of IDR[31]. The S120, S200 serine residues which can be phosphorylated[22], and the R192 arginine residue which is associated with Parkinson's disease[23], are all

**Fig. 2 | rRNA was essential to maintain the localization and LLPS of LIN28A and LLPS of nucleolar protein FBL in mouse ESC. a** Confocal microscopy Airyscan imaging of the eGFP-LIN28A protein expression in transduced control, CX-5461-treated, Pol I degraded and snoRNA knockout mESCs. Scale bar, 10 μm. **b** Upper panel: FRAP analysis showing eGFP-LIN28A recovery after photobleaching in control and snoRNA knockout cells. $n = 3$ biologically independent experiments. Nu: targeted regions in the nucleus. Lower panel: FRAP analysis showing eGFP-LIN28A recovery after photobleaching in control, CX-5461-treated, Pol I degraded, and snoRNA knockout cells. Cyto: targeted regions in the cytoplasm. Data are presented as mean values +/− SEM. **c** Representative FRAP images of eGFP-LIN28A in control, CX-5461-treated, Pol I degraded, and snoRNA knockout ESCs. The targeted bleached region is highlighted in a white circle. Scale bar, 5 μm. **d** Statistical analysis of LIN28A protein loci numbers in the nucleus in the above cells; $n = 33$ cells. One-way ANOVA. **e** Statistical analysis of LIN28A nucleus/cytoplasm fluorescence intensity ratios in the above cells; $n = 24$ cells. One-way ANOVA. **f** Statistical analysis of LIN28A loci area in the nucleus in the above cells; $n = 21$ cells. One-way ANOVA. **g** Schematic representation of the eGFP-LIN28A/FBL-mCherry mESC line generated. **h** Confocal microscopy Airyscan images of the morphology and nucleolar localization of LIN28A and FBL in the living control, CX-5461-treated, Pol I degraded, and snoRNA knockout mESCs. Scale bar, 5 μm. **i** Statistical analysis of the numbers the typical morphology of FBL in the living control, CX-5461-treated, Pol I degraded, and snoRNA knockout ESCs; $n = 20$ nucleoli. Fisher Exact Test, two-sided; Ctrl vs CX-5461:$p = 3.35E-09$; Ctrl vs de Pol I:$p = 3.35E-09$, Ctrl vs snoRNA KO:$p = 3.36E-06$. For (**a**, **c**, **h**) three times biologically independent experiments were performed. For (**d**–**f**), the center line is the median, the bottom of the box is the 25th percentile boundary, the top of the box is the 75th, percentile, and the top and bottom of the vertical line define the boundary of the data. Source data are provided as a Source Data file.

in the IDRs of LIN28A and conserved between humans and mice (Fig. 4a). Several studies showed that serine phosphorylation influenced phase separation[31,32]. To gain more insight into the role of these amino acids in LIN28A phase separation, we engineered stable E14 ESC lines expressing phospho-null (both S120A and S200A) Mut LIN28A, in which serine phosphorylation was abrogated by substitution with alanine (Fig. 4b). Our FRAP analysis showed that phospho-null Mut LIN28A had lower fluidity in the nucleoli (Fig. 4c), with a simultaneous lower fluidity of FBL (Fig. 4k, l). Then, we separately engineered stable E14 ESC lines expressing S120 phospho-null LIN28A or S200 phospho-null LIN28A (Fig. 4b). FRAP analysis showed that both single mutation phospho-null LIN28A proteins showed lower mobility in the nucleolus (Fig. 4d, f), with a simultaneous lower fluidity of FBL in the nucleolus (Fig. 4k, l).

Recently, a loss-of-function variant of LIN28A (R192G substitution) was identified in two early-onset PD patients, and the mutation led to developmental defects and PD-related phenotypes in midbrain dopamine neurons without a known mechanism in LLPS[23]. Accordingly, wild type LIN28A overexpression was reported to promote the therapeutic potential of cultured neural stem cells in a Parkinson's disease model[23]. To test the effect of R192 mutation on LIN28A phase separation, we engineered stable E14 ESC lines expressing a LIN28A mutant, in which R192 was mutated to glycine (R192G) (Fig. 4b). FRAP analysis showed that the R192G single mutated LIN28A protein had lower fluidity in the nucleolus (Fig. 4d, f), and so as the FBL protein (Fig. 4k, l). Moreover, in contrast to wild type LIN28A in the nucleoli with a round and porous morphology surrounding FBL, all the LIN28A mutants showed hollowed loose loop structures, and the distribution of FBL was no longer constrained by LIN28A and appeared to have a diffused distribution within the LIN28A shell (Fig. 4h). We already showed that cytoplasmic LIN28A did not have typical phase-separated condensate behavior. It is worth noting that the amino acids mutation in the LIN28A' IDR region did not affect its distribution and fluidity in the cytoplasm (Fig. 4e, g, h).

To confirm the nucleolar disruptive role of LIN28A mutants was resulted from their weakened phase separation property, we generated the rescuing LIN28A mutants by fusing the exogenous IDR of FUS (S120A-FUS IDR, S200A-FUS IDR, R192G-FUS IDR), which is known to drive phase separation[32–34] (Fig. 4b). Notably, the fused IDRs rescued the morphology and phase separation capability of the LIN28A mutants (Fig. 4d, f, h). Meanwhile, the three IDR fusions completely rescued the impaired fluidity of FBL caused by the LIN28A mutants in the nucleolus (Fig. 4k, l). The statistical analysis of the numbers of the typical representative morphology of FBL in living WT, *Lin28a* KO, and *Lin28a* KO cells transduced with Mut-LIN28A, S120A LIN28A, S200A LIN28A, R192G LIN28A, and FUS protein's IDR-fused LIN28A variants quantitatively showed that the key amino acids mutations at the LIN28A'IDR region impaired nucleolar phase separation(Fig. 4i).

Meanwhile, we found that the nucleolar LIN28A morphology in these key amino acids mutants were more similar to WT cells cultured in LIF/2i medium, whereas the FUS protein's IDR fused LIN28A were more similar to WT cells cultured in LIF/serum medium or the primed state medium with FGF2/Activin A(Fig. 4h, n). Statistical analysis of the numbers of the representative morphology of LIN28A quantitatively reflected this trend (Fig. 4j, o). This suggests that LIN28A phase separation in the nucleolus may play a role in the pluripotency transition of mouse embryonic stem cells. This drove us to consider the link between LIN28A-regulated nucleolar phase separation and pluripotency state transition in mouse embryonic stem cells.

Together, these results demonstrated that the key amino acids at the IDR regions were essential for LIN28A phase separation, as well as for maintaining normal nucleolar phase separation(Fig. 4m), and the phase separation property of LIN28A might be involved in its role in regulating pluripotency.

To investigate whether these IDR mutations can influence LIN28A's canonical function of microRNA let-7 binding, the WT, S120A, R192G, S200A, and F47A point mutations mouse recombinant LIN28A proteins were prepared for electrophoretic mobility shift assay (EMSA) (Supplementary Fig. 6a). Mouse let-7 family miRNAs comprise 12 members, the mature miRNA sequence of which is highly conserved between the different genes. The LIN28A F47 located at the CSD domain has been identified as a single amino acid residue required for binding to pre-let-7g[35], and the F47A point mutation abolished LIN28A binding to pre-let-7. In contrast, we found that the S120A, R192G, and S200A LIN28A IDR point mutations had the similar capacity for binding to pre-let-7g as WT LIN28A (Supplementary Fig. 6c, d). qPCR analysis also showed that S120, S200 and R192 can be mutated without affecting the expression of mature let-7g(Supplementary Fig. 6b). These data suggested that the IDR mutations did no influence LIN28A's function in let-7 binding. Yet it is worth noting that our study does not completely rule out that there are other RNAs that can be affected by the S120A, S200A and R192G mutations.

**The phase separation property of LIN28A is required for its role in primed pluripotency state conversion in mouse embryonic stem cells and reprogramming of somatic cells**

Regarding the two pluripotent states: the naive state represents the pre-implantation inner cell mass, and the primed state represents the post-implantation epiblast cells in the murine early embryo development[2]. Wild type mouse ESC cultured in FGF2/Activin A medium express higher primed state marker genes such as *Otx2* and *Fgf5*, and lower naive state marker genes such as *Klf4*, *Nanog*, and *Esrrb*, compared with cultured in the LIF/2i condition. *Lin28a* knockout cells cultured in FGF2/Activin A medium showed a robust delay of naïve marker repression, and primed marker induction compared to wild type cells, confirming that normal LIN28A promoted mouse ESC conversion to a primed state[5,7]. When we overexpressed full length wild

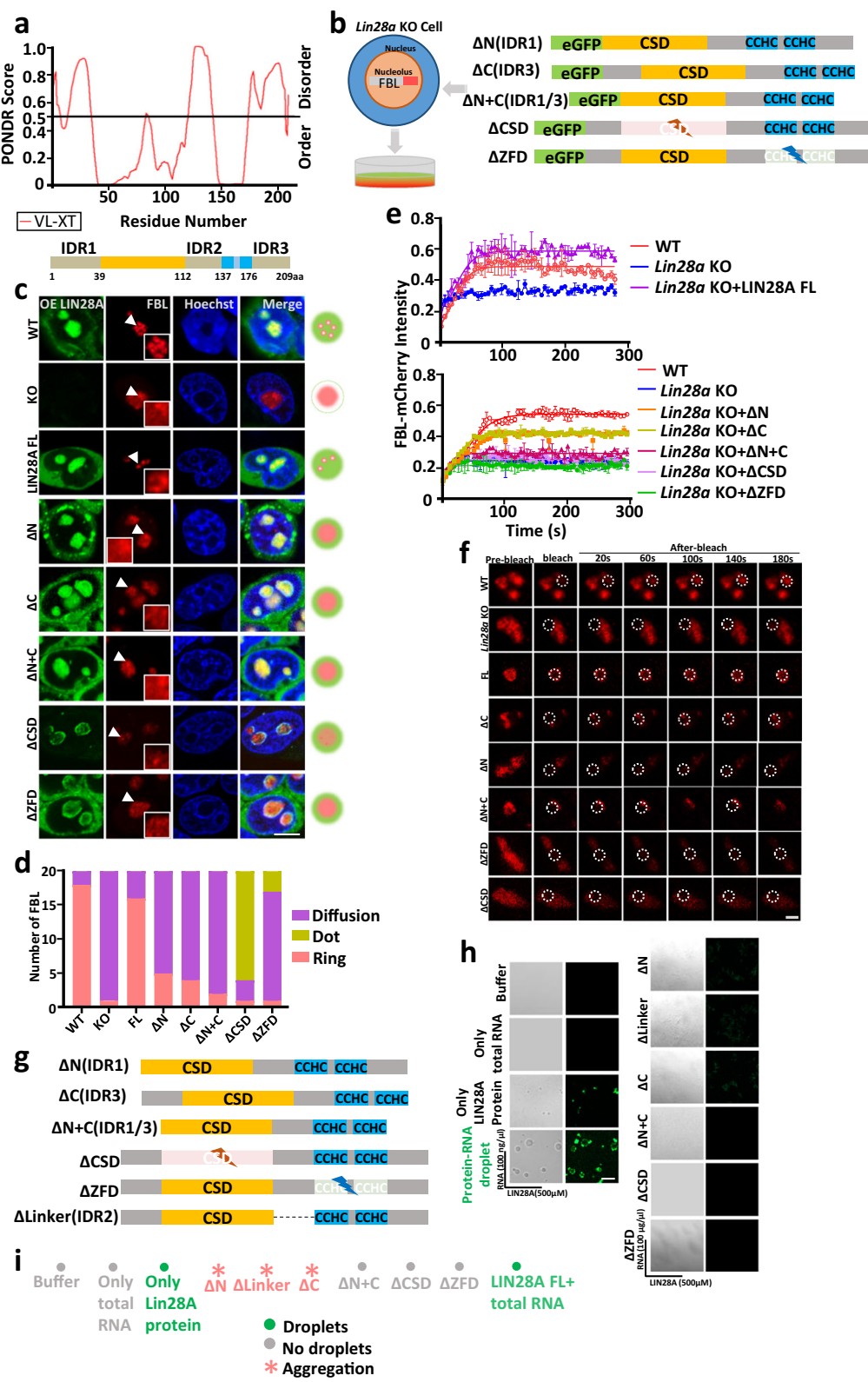

type (WT) LIN28A and phospho-null LIN28A mutants (Mut LIN28A) in LIN28A knockout cells, wild type LIN28A rescued the conversion to the primed state. Strikingly, double mutant LIN28A (both S120A and S200A) was unable to promote ESC exit from the naïve state, as NANOG immunostaining showed that Mut LIN28A tended to maintain the naïve state marker (Fig. 5a, b). qPCR analysis also showed that Mut LIN28A cells cultured in FGF2/Activin A medium showed an impaired

decrease in *Nanog* and *Klf4*, and impaired increase in *Fgf5* and *Otx2* compared with WT LIN28A mESC (Supplementary Fig. 5a, b). In fact, all three S120A, S200A and R192G single mutation LIN28A lost the function of promoting ESC exit from the naïve state, and their IDR fusions completely rescued this impairment indicated by NANOG immunostaining (Fig. 5c, d). qPCR analysis also showed that LIN28A mutant IDR fusions, like WT LIN28A, enabled ESCs to complete the naive-to-

**Fig. 3 | LIN28-RNA forms LLPS in an RBD and IDR domain-dependent manner in vivo and in vitro. a** LIN28A intrinsically disordered regions were predicted by PONDR (Predictor of Natural Disordered Regions; http://pondr.com/). **b** A schematic diagram of the constructs used to investigate the function of individual domains of LIN28A in FBL-mCherry expressing stable *Lin28a* KO cells. **c** Representative confocal microscopy Airyscan images of the morphology and nucleolar localization of LIN28A and FBL in living WT, *Lin28a* KO, LIN28A full-length LIN28A overexpressing, and truncated LIN28A overexpressing cells. Scale bar, 5 μm. **d** Statistical analysis of the numbers the typical morphology of FBL in the WT, *Lin28a* KO, LIN28A full-length LIN28A overexpressing, and truncated LIN28A overexpressing cells; *n* = 20 nucleoli. Fisher Exact Test, two-sided; WT vs KO:*p* = 5.82E-08, WT vs FL:*p* = 0.6614197, WT vs ΔN:*p* = 6.86063E-05, WT vs ΔC:*p* = 1.66E-05, WT vs ΔN + C:*p* = 5.30E-07, WT vs ΔCSD:*p* = 5.82E-08, WT vs ΔZFD:*p* = 1.95E-07.

**e** FRAP analysis showing FBL-mCherry recovery after photobleaching in the above cells. *n* = 3 biologically independent experiments. Data are presented as mean values +/− SEM. WT curves shown in two quantification graphs were derived from independent experiments. **f** Representative FRAP images of FBL in living WT, *Lin28a* KO, and *Lin28a* KO mESCs transduced with full length WT LIN28A or individual domain deleted LIN28A variants. Scale bar, 5 μm. **g** Constructs used to investigate the function of individual domains of LIN28A in vitro. **h** LLPS of purified recombinant LIN28A protein in 50 mM NaCl and 100 ng/μL total RNA. Scale bar, 10 μm. **i** Summary of LLPS of purified recombinant LIN28A protein under indicated conditions, in the presence of 50 mM NaCl in vitro. For (**c**, **f**, **h**) three times biologically independent experiments were performed. Source data are provided as a Source Data file.

primed state conversion, indicated by robust *Nanog* repression and *Otx2* induction (Supplementary Fig. 5c). These results demonstrated that it was the nucleolar phase separation property that could mediate the naive-to-primed state conversion.

We also compared the nucleolar features and the translation function in both the wild type and *Lin28a* KO mESC in naive or primed states. The STED imaging demonstrated the DFC component FBL was embedded and immersed within the granular component NPM in the wild type mESC. FBL formed the typical 'wreath' structure in primed wild type mESC indicating more developed DFC units in the primed state (Fig. 5e, f). We next assessed the DFC and GC assembly in *Lin28a* KO cells in response to naive-to-primed state conversion, and found that loss of LIN28A resulted in blurry stratification of nucleoli. This abnormal stratification of nucleoli did not improve during naive-to-primed state conversion (Fig. 5e). To gain more spatial details of FBL in the DFC cluster, we further analyzed the three-dimensional Z-stack STED images of FBL and found that FBL exhibited clear cluster-like patterns[28,29]. The DFCs in the wild type naive mESCs consisted of 3-4 clusters on average, and 5–6 clusters for the primed state (Fig. 5f–h). *Lin28a* KO led to reduced volume and number of FBL clusters in both states, and particularly larger margin of reduction in the primed state (Fig. 5f–h), demonstrating LIN28A's role in supporting DFC during the primed state conversion.

By FRAP analysis, we also found the fluidity of nucleolar LIN28A and FBL were faster in the primed state compared with the naive state (Supplementary Fig. 7c, d). Considering the function of the nucleolus in regulating ribosome and translation, we determined the protein synthesis rate in the naïve and primed cells using OP-Puromycin staining. Consistent with our above view, we found that the primed state cells had significantly enhanced protein synthesis. However, the efficiency of protein synthesis was abolished in *Lin28a* knockout cells in the FGF2/Activin A primed state medium (Supplementary Fig. 7a, b). Since the naïve and primed cells in vitro corresponded to pre- and post-implantation epiblast cells. Using published RNA-seq data[36], we also compared the expression of nucleolus, translation, and ribosome genes in E4.5 ICM and E6.5 post-implantation epiblast. We found that the nucleolus, translation, and ribosome genes were more highly expressed in the E6.5 post-implantation epiblast cells (Supplementary Fig. 7e). Together, these results demonstrated that the nucleolar morphology and functions became more matured during the naive-to-primed conversion.

As LIN28A is a reprogramming factor, and its loss leads to reduced reprogramming efficiency[5,7], we wonder what is the role of the nucleolar phase separation property of LIN28A in the reprogramming process. Therefore, we generated mouse and human LIN28A mutants and IDR fusions (Fig. 7a). For the mouse embryonic fibroblast (MEF) cells and neonatal human dermal fibroblast (NHDF) cells reprogrammed by OCT4, SOX2, NANOG and LIN28A, LIN28A IDR point mutations led to reduced reprogramming efficiency, which can be rescued by fusion of FUS IDRs (Figs. 6a, b, 7b, c), demonstrating phase separation of LIN28A facilitated reprogramming.

We further characterized the mouse iPSCs. Stimulated Emission Depletion Microscopy (STED) imaging revealed significant differences in the morphology and stratification of the nucleolus between MEF cells and iPSCs. In the MEF cells, NPM was in irregular shape, whereas in the iPSCs, NPM showed the round 'lotus root' structure and was colocalized with LIN28A (Fig. 6c). We further quantified the regularity of the granular component (GC) using Boyce-Clark semidiameter index which was originally used to assess the 'compactness' of space layouts[37]. GC showed higher degree of regularity in iPSCs compared with MEF (Fig. 6e). Besides, the STED imaging showed the DFC component FBL tended to show the 'ring' structure, and it was embedded and immersed within the granular component NPM in the iPS cells indicating more developed DFC units and clearly stratified nucleoli (Fig. 6d–g). OP-Puromycin staining indicated that iPSCs possessed higher protein synthesis rate compared with MEF cells, suggesting the more clearly stratified nucleoli in iPSCs were functionally more developed (Fig. 6h). Taken together, these results demonstrated that the IDR region of LIN28A that regulated phase separation played an important role in reprogramming of mouse iPSCs.

Stimulated Emission Depletion Microscopy (STED) imaging also revealed significant differences in the morphology and stratification of the nucleoli between NHDF cells and iPSCs. FBL showed the round 'wreath' structure indicating more developed DFC units and was immersed within LIN28A in the iPSCs (Fig. 7d, e). FBL showed higher degree of regularity in iPSCs compared with NHDF (Fig. 7g). GC layer marked by NPM displayed a 'ring' structure in the iPSCs, whereas in the NHDF cells it assumed an irregular shape (Fig. 7f). OP-Puromycin staining indicated that iPSCs possessed higher protein synthesis rate compared with NHDF cells (Fig. 7h, i), suggesting the more clearly stratified nucleoli with more DFC clusters in iPSCs were functionally more developed.

Together, these results demonstrated that the induction of LIN28A during reprogramming contributed to phase stratification of nucleoli and promoted the translation function of nucleoli in the induced pluripotent stem cells, and the non-canonical phase separation property of LIN28A mediated these nucleolar remodeling processes and the subsequent cell fate decisions (Supplementary Fig. 9a).

Finally, we also examined whether the phase separation property of LIN28A is necessary for its role in differentiation of mouse pluripotent stem cells. The undifferentiated state of mouse ES cells is maintained in the presence of leukemia inhibitory factor (LIF) in the culture medium[38–40], whereas the spontaneous differentiation of ES cells can be triggered by withdrawal of LIF from the medium[41,42]. qPCR analysis showed that the expression of *Lin28a* decreased sharply after LIF withdrawal in 3 days (Supplementary Fig. 8a). The IDR mutated LIN28A variants slightly delayed the differentiation at day 3, but had no change at day 5 and day 7 (Supplementary Fig. 8b). The possible reason is that these mutants had the tendency to stay in a state closer to the naive state, and had slower kinetics to exit the pluripotency as illustrated in the Fig. 5c, d above. As LIN28A is decreased sharply during differentiation, we expect that it does not have a role in regulating

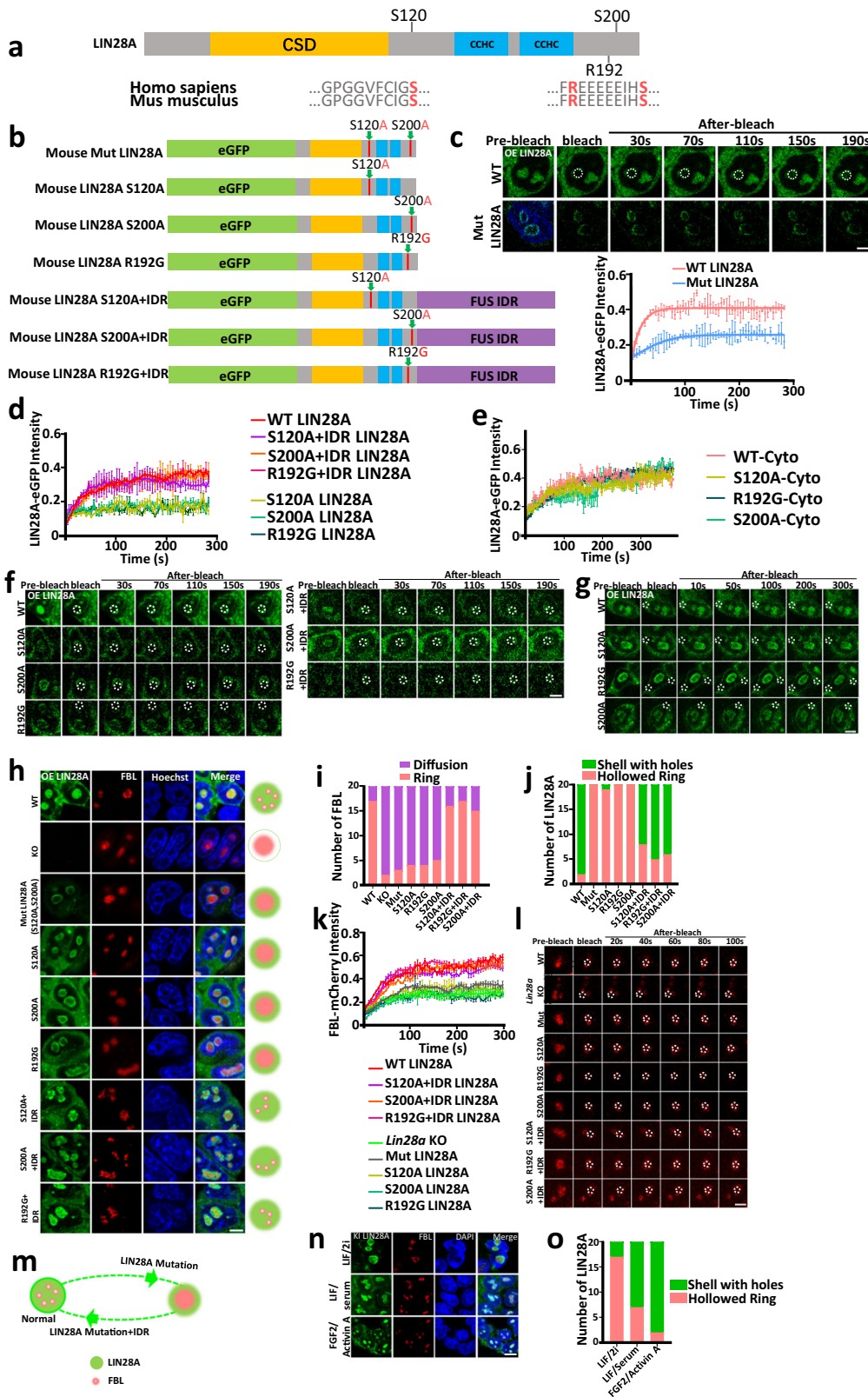

differentiation genes. Our qPCR analysis also showed that the IDR mutant LIN28A did not affect the expression kinetics of the marker genes of the three germ layers (Supplementary Fig. 8c–e).

## Discussion

The reprogramming process and the naive versus primed state of pluripotency have been intensively studied in terms of their

transcriptional regulation, epigenetic remodeling, and metabolic reprogramming[2,7]. However, an important organelle related to translation and stress response, the nucleolus, has not been characterized in this process. Also, as a typical example of membrane-less and compartmentalized condensate, the LLPS of nucleolus in this process has not been studies as well. We have revealed the contrasting increased number of nucleoli and DFC units, translational activity, as well as

**Fig. 4 | Key amino acids at the IDR region are essential for LIN28A and nucleolar phase separation. a** Schematic illustration of human and mouse LIN28A domains. **b** Constructs of individual mutation of mouse LIN28A. **c** FRAP analysis showing WT-LIN28A and Mut-LIN28A recovery after photobleaching in the nucleolus. Shown representative images. Scale bar, 5 μm. **d** FRAP analysis showing WT-LIN28A, S120A, S200A, R192G, and IDR fusions recovery after photobleaching in the nucleolus. **e** FRAP analysis showing WT-LIN28A, S120A, S200A, and R192G variants recovery after photobleaching in the cytoplasm. **f** Representative images of fluorescence recovery after photobleaching (FRAP) of LIN28A variants in (**d**) in the nucleolus. Scale bar, 5 μm. **g** Representative images of fluorescence recovery after photobleaching (FRAP) of LIN28A variants in (**e**) in the cytoplasm. Scale bar, 5 μm. **h** Representative confocal microscopy Airyscan images of the morphology and nucleolar localization of LIN28A and FBL in living WT, *Lin28a* KO, and *Lin28a* KO cells transduced with Mut-LIN28A, S120A LIN28A, S200A LIN28A, R192G LIN28A, and FUS protein's IDR fused LIN28A variants in LIF/Serum medium. Scale bar, 5 μm. **i** Statistical analysis of the numbers of the typical morphology of FBL in (**h**); *n* = 20 nucleoli. Fisher Exact Test, two-sided; WT vs KO:*p* = 3.35795E-06, WT vs Mut:*p* = 1.93853E-05, WT vs S120A:*p* = 8.75018E-05, WT vs R192G:*p* = 8.75018E-05, WT vs S200A:*p* = 0.000328419, WT vs S120A + IDR:*p* = 1, WT vs R192G + IDR:*p* = 1, WT vs S200A + IDR:*p* = 0.6947647. **j** Statistical analysis of the numbers the typical morphology of LIN28A in living WT, and *Lin28a* KO cells transduced with Mut-LIN28A, S120A-LIN28A, S200A-LIN28A, R192G-LIN28A, and FUS protein's IDR fused LIN28A variants; *n* = 20 nucleoli. Fisher Exact Test, two-sided; WT vs KO:*p* = 3.35155E-09, WT vs S120A:*p* = 5.81952E-08, WT vs R192G:*p* = 3.35155E-09, WT vs S200A:*p* = 3.35155E-09, WT vs S120A + IDR:*p* = 0.06483316, WT vs R192G + IDR:*p* = 0.4074844, WT vs S200A + IDR:p = 0.2351162. **k** FRAP analysis showing FBL-mCherry recovery after photobleaching in (**h**). **l** Representative FRAP images of FBL in (**h**). Scale bar, 5 μm. **m** A cartoon diagram showing morphological changes of LIN28A and FBL. **n** Confocal microscopy Airyscan images of the morphology and Localization of LIN28A and FBL in eGFP-LIN28A knock-in E14 mESCs cultured in LIF/2i(Naïve), LIF/Serum, FGF2/Activin A(Primed) medium. Scale bar, 5 μm. **o** Statistical analysis of the numbers the typical morphology of LIN28A in eGFP-LIN28A knock-in E14 mESCs cultured in LIF/2i(Naïve), LIF/Serum, FGF2/Activin A(Primed) medium; *n* = 20 nucleoli. Fisher Exact Test, two-sided; Naïve vs LIF/Serum:*p* = 0.003056449, Naïve vs Primed:*p* = 3.35795E-06.For **c**–**h**, **k**, **l**, **n**, three times biologically independent experiments were performed. For **c**, **d**, **e**, **k**, data are presented as mean values +/− SEM. Source data are provided as a Source Data file.

fluidity of the components of the compartmentalized condensate during the naive-to-primed state transition and the reprogramming. LIN28A mediates these processes through its non-canonical function related to the LLPS-promoting properties.

The pluripotent factor LIN28A is a highly conserved RNA binding protein[43]. LIN28A contains two well-known RNA-binding domains (RBDs), a cold-shock domains (CSD), and a cysteine cysteine histidine cysteine (CCHC) zinc-finger domains (ZFD). A flurry of studies show that LIN28A has an important role in reprogramming and maintenance of pluripotency through *let-7* dependent mechanisms based on its RBDs. We made the truncated mutants of RBDs and IDRs, and found that both RBDs and IDRs of LIN28A were important for proper organization of nucleolus. Almost all previous studies focused on its two RBDs, and on seeking the RNA binding targets to elucidate the mechanistic roles of LIN28A in regulating development[9,44–46], cell fate reprogramming[6,11,47], and cellular and whole body metabolism[7,43,48]. Other regions in LIN28A, especially the intrinsically disordered regions, due to their highly unpredictable structure, have been largely neglected, or considered to have no functions[49,50]. We were more curious about the function of IDRs, which was previously assumed to have no functional roles in LIN28A. Therefore, in this article, we focus on IDRs which was rarely studied before.

Both RNA binding domains and IDRs can contribute to phase-separated condensate formation. Our study here dissected each of their contribution by making specific truncations and mutations in the RBDs and IDRs. Strikingly, three single point mutations at the IDR regions can each individually disrupt the phase separation propensity of LIN28A itself and its associated nucleolar proteins, illustrating a clear role of the IDRs in mediating the phase separation of LIN28A and in maintaining the nucleolar integrity.

Traditionally, the role of LIN28A in cell fate decision is mainly attributed by its cytoplasmic functions in binding and repressing microRNA *let-7*[10,11,21,49–51], or binding mRNAs to have post-transcriptional regulation[52]. Our study here illustrated that a majority of LIN28A protein plays its role in the nucleolus in ESCs, and its function is conferred through both RNA binding-dependent (in this case, nucleolar RNA such as rRNA), and RNA binding-independent phase separation mechanisms. We also revealed the nucleolar function and phase separation mechanisms of LIN28A in reprogramming and in the transition of naive-to-primed pluripotency states. Phospho-null LIN28A (S120A, S200A, and S120A&S200A double mutations) might alter the charge and hydrophobicity of IDRs, and subsequently led to impaired fluidity of phospho-null LIN28A mutants. Strikingly, just a single IDR region mutation was able to abolish LIN28A's capacity to promote the naive state exit and the primed state transition, as well as the reprogramming efficiency.

Finally, it has been recently proposed that disease-associated mutations are prevalently located in the IDRs, and they have been frequently overlooked or annotated as variants of unknown significance[53–55]. We specifically examined a recently reported LIN28A variant R192G located in the IDR region in the C-terminal associated with parkinson's disease without a known mechanism[23]. Our data support a role of this mutation in influencing the phase separation propensity of LIN28A. More work related to dissect this role in a neuronal system and to therapeutically target phase separation of LIN28A warrants further investigations.

## Methods
### Cell culture
Mouse (male) E14Tg2A (E14) ESCs (a gift from George Q. Daley's lab (Harvard Medical School) (ATCC, CRL-1821)) were used for almost all experiments. E14 *Lin28a* KO cells were constructed as our previously described[19]. E14 eGFP knock-in cells and E14 cells stably expressing eGFP- LIN28A/eGFP-LIN28A truncations/eGFP-LIN28A mutations/NCL-eGFP/FBL-mCherry were constructed in our study and used in corresponding experiments. The snoRNA knockout ESC lines (the homologs of human SNORD113-114 gene cluster was successfully knocked-out) was a gift from Pengxu Qian's lab (Zhejiang university). The Pol I-degraded ESC lines was a gift from Xiong Ji's lab (Peking University). mESCs were cultured on 0.1% gelatin-coated plates with MEF feeder cells in ES-FBS culture medium (DMEM, 1967762, Gibco), 15% fetal bovine serum (FBS, 10099-141, Gibco) and 1000 U/ml LIF (PEPRO TECH), 0.1 mM nonessential amino acids (GNM71450, GENOM), 100 U/ml penicillin, 100 μg/ml streptomycin (15140-122, Gibco) and 0.1 mM 2-Mercaptoethanol (Millipore). For the naive state 2i culture media, 1 μM PD0325901 and 3 μM CHIR99021 (STEMCELL Technologies) were supplemented into a 1:1 mix of DMEM/F12 (11320-033, Gibco) and Neurobasal medium (21103-049, Gibco) containing N2 and B27 supplements (1:100 dilutions of 17502-048 and 17504-044, Life Technologies), 5000 U/ml penicillin and streptomycin(15140-122, Gibco), 0.1 mM nonessential amino acids(GNM71450, GENOM), 0.1 mM 2-mercaptoethanol, 1000 U/ml LIF(PEPRO TECH). For the primed pluripotency state transition, 20 ng/mL Activin A, 10 ng/mL FGF2, and 1% KSR were supplemented to the 1:1 DMEM/F12 and Neurobasal medium containing N2 and B27. The MEF cell line (established in Jin Zhang's lab) was cultured in Dulbecco's modified eagle medium (DMEM, 1967762, Gibco) supplemented with 10% FBS (10099-141,

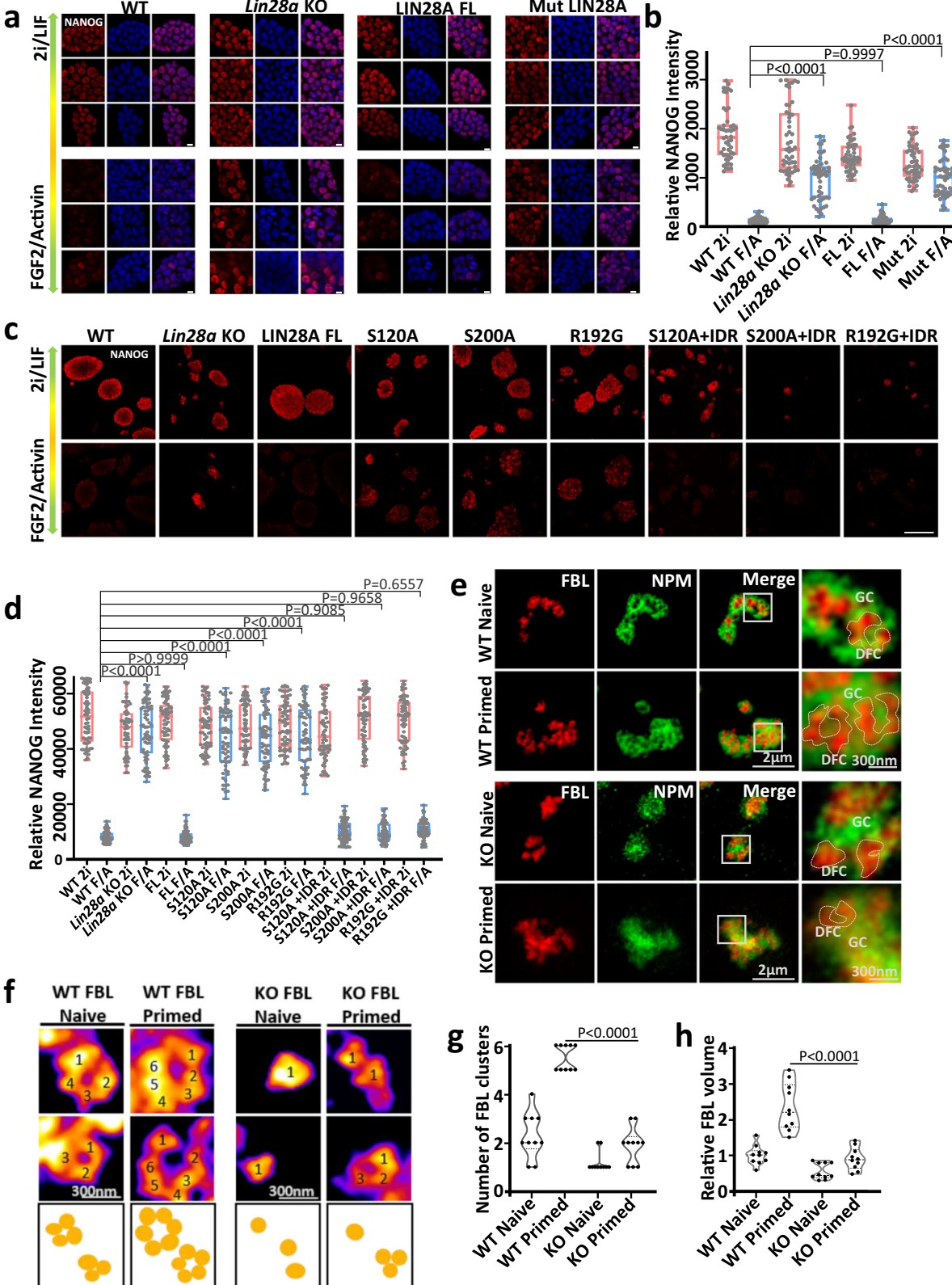

Gibco) supplemented with 100 U/ml penicillin and 100 μg/ml streptomycin (15140-122, Gibco). All cells were grown at 37∘C with 5% CO2.

**Constructs**

DNA fragments encoding mouse LIN28A, FBL, and NCL were PCR-amplified from the E14 cell cDNA library. The cDNA library was created using the HiScript II Q RT SuperMix System (R223-01, Vazyme). DNA fragments encoding LIN28A were inserted into the pSIN-eGFP-puro or Lenti-mCherry-puro backbone, FBL was inserted into the Lenti-mCherry-blasticidin backbone, and NCL was inserted into the Lenti-eGFP-puro backbone. DNA fragments encoding mouse S120A LIN28A, R192G LIN28A, S200A LIN28A, S120A-S200A LIN28A, and three IDR fusions were inserted into the pSIN-eGFP-puro backbone.DNA fragments encoding human S120A LIN28A, R192G LIN28A, S200A LIN28A, and three IDR fusions were also inserted into the pSIN-eGFP-puro backbone.

**Fig. 5 | The key amino acids mediating LLPS of LIN28A is required for its role in naïve-to-primed pluripotency conversion of mouse ESCs. a** NANOG immunostaining of WT, *Lin28a* KO, and *Lin28a* KO mESCs transduced with full length WT LIN28A or Mut-LIN28A converted from the naïve state to the primed state. Scale bar, 10 μm. **b** Statistical analysis of NANOG protein fluorescence intensity of the above cells in (**a**). n = 64 cells. One-way ANOVA. **c** NANOG immunostaining of WT, *Lin28a* KO, and *Lin28a* KO mESCs transduced with full length WT LIN28A, single mutation variants, or IDR-fused variants, converted from the naïve state to the primed state. Scale bar, 200 μm. **d** Statistical analysis of NANOG protein fluorescence intensity of the above cells in (**c**). n = 64 cells. One-way ANOVA. **e** Representative STED immunofluorescence images of nucleoli in WT and *Lin28a*

KO mESCs in the naïve state and the converted primed state. FBL applied STED pattern. **f** Typical FBL in WT and *Lin28a* KO mESCs in (**e**) by STED. *Lin28a* KO led to decreased size of FBL rings compared with WT when ESCs were converted to the primed state. Scale bar, 300 nm. **g** Statistical analysis of the number of FBL clusters in the above conditions. n = 10 nucleoli. Two-way ANOVA. **h** Statistical analysis of the relative FBL volume in the above conditions. n = 10 nucleoli. Two-way ANOVA. For (**b**, **d**) the center line is the median, the bottom of the box is the 25th percentile boundary, the top of the box is the 75th percentile, and the top and bottom of the vertical line define the boundary of the data. Source data are provided as a Source Data file.

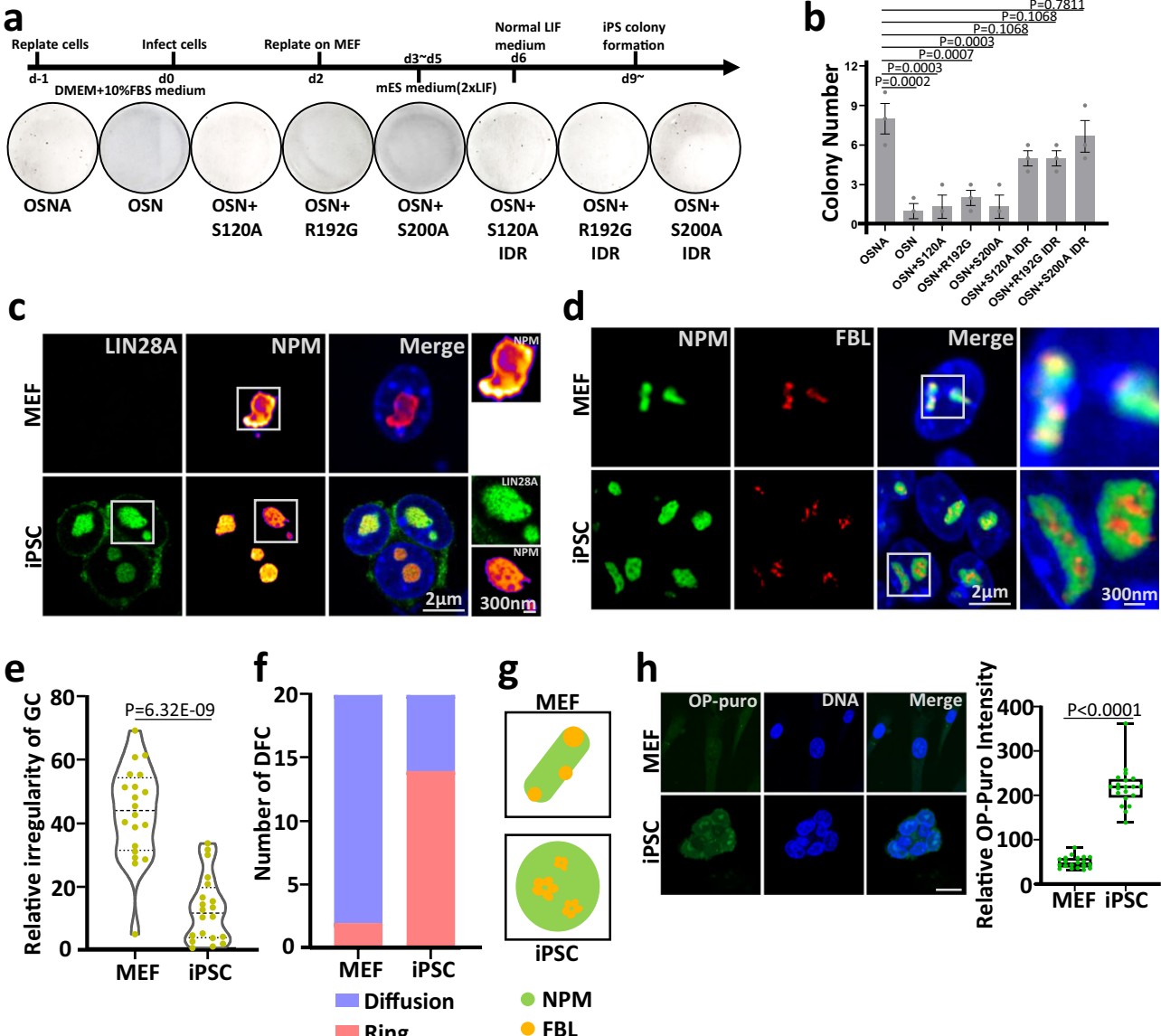

**Fig. 6 | The key amino acids mediating LLPS of LIN28A is required for its role in facilitating reprogramming of mouse embryonic fibroblast cells(MEF).**
**a** Reprogramming efficiency of MEF cells transduced with OCT4, SOX2, NANOG, and LIN28A variants. **b** Number of iPSC colonies 14 days after OSNA transduction of MEF cells. n = 3 biologically independent experiments. One-way ANOVA. Data are presented as mean values +/− SEM. **c** Representative STED immunofluorescence images of LIN28A and NPM in MEF or iPSCs. NPM applied STED pattern.
**d** Representative STED immunofluorescence images of NPM and FBL in MEF or iPSCs. FBL applied STED pattern. **e** Statistical analysis of the GC(NPM) irregularity in MEF and iPSCs using Boyce-Clark semidiameter index. The larger the number, the

more irregular the NPM. n = 20 nucleoli, unpaired two-tailed student's *t*-test.
**f** Graph showing the number of typical DFC(FBL) morphology in MEF or iPSCs. n = 20 nucleoli. Fisher Exact Test, two-sided; MEF vs iPSC:p = 0.000244362. **g** A cartoon diagram showing morphological changes of NPM and FBL in MEF and iPSCs. **h** Images and OP-Puro intensity statistical analysis of OP-puromycin-labeled MEF and iPSCs. Scale bar, 20 μm. n = 20 cells, unpaired two-tailed student's *t*-test. The center line is the median, the bottom of the box is the 25th percentile boundary, the top of the box is the 75th percentile, and the top and bottom of the vertical line define the boundary of the data. Source data are provided as a Source Data file.

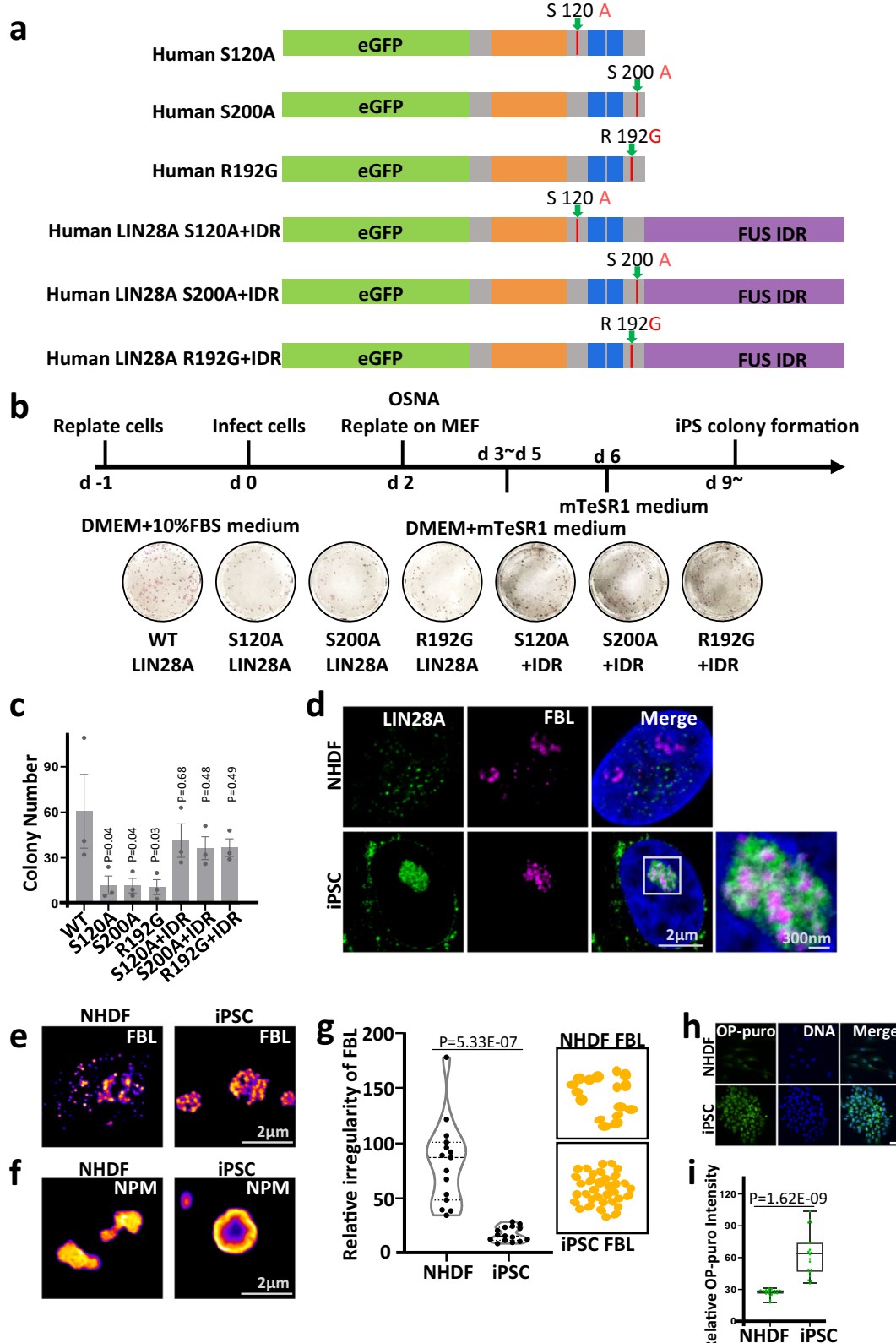

### Generation of *Lin28a* knockout ESC lines with CRISPR/Cas9

sgRNAs were designed to target the second exon of *Lin28a* using the online tool: (https://portals.broadinstitute.org/gpp/public/analysistools/sgrna-design), then the sgRNA was inserted into the gRNA-Cas9-Puro backbone (L00691, GenScript). ESCs were nucleofected with the plasmid containing the sgRNA and Cas9 using Lonza 4D Nucleofector. 48 h after transfection, ESCs were selected with 1 µg/mL puromycin for

7 days. Clones were picked and LIN28A expression was detected by Western blotting. The sgRNA used is listed in Supplementary Table 2.

### Generation of eGFP-*Lin28a* knock-in ESC lines with CRISPR/Cas9

sgRNAs were designed to target the *Lin28a* N-terminal genomic loci using the online tool: (https://portals.broadinstitute.org/gpp/public/analysistools/sgrna-design), then the sgRNA was inserted into the

**Fig. 7 | The key amino acids mediating LLPS of LIN28A is required for its role in facilitating reprogramming of human fibroblast cells. a** Constructs used to investigate the function of mutations in IDRs of human LIN28A. **b** Reprogramming efficiency of NHDF cells transduced with human OCT4, SOX2, NANOG, and LIN28A variants in (A). **c** Number of iPSC colonies 21 days after OSNA transduction of NHDF cells. *n* = 3 biologically independent experiments. One-way ANOVA. Data are presented as mean values +/− SEM. **d** Representative STED immunofluorescence images of LIN28A and FBL in NHDF or iPSCs. FBL applied STED pattern. **e** FBL in NHDF and iPSC cells by STED. FBL exhibited a wreath-like structure in iPSCs. **f** NPM in NHDF and iPSC cells by STED. NPM exhibited a ring-like structure in iPSCs.

**g** Statistical analysis of the FBL regularity in NHDF and iPSCs using Boyce-Clark semidiameter index. The larger the number, the more irregular the FBL. *n* = 15 nucleoli, unpaired two-tailed student's *t*-test. **h** Immunofluorescence imaging showing OP-puromycin-labeled NHDF and iPSCs. Scale bar, 100 μm. **i** OP-Puro intensity statistical analysis of OP-puromycin-labeled NHDF and iPSC cells. *n* = 20 cells, unpaired two-tailed student's *t*-test. The center line is the median, the bottom of the box is the 25th percentile boundary, the top of the box is the 75th percentile, and the top and bottom of the vertical line define the boundary of the data. Source data are provided as a Source Data file.

PX459-gRNA-Cas9-Puro backbone. The donor plasmid contained eGFP sequence with left and right homology arms. ESCs were transfected with the donor plasmid and PX459-gRNA-Cas9-Puro plasmid using Lipo2000 Transfection Reagent. 48 h after transfection, ESCs were selected with 1 μg/ml puromycin for 7 days. Clones were picked and the knock-in insertion was detected by PCR. The sgRNA used is listed in Supplementary Table 2.

## Western blotting
Related ESCs were lysed with RIPA lysis buffer containing protease-inhibitors for 30 min on ice and centrifuged at 13201 ×g for 15 min, and the supernatant was carefully transferred to new tubes. Protein concentration was quantified using the BCA protein assay kit (P0012, Beyotime), and 40 μg of denatured protein samples were separated by 10% SDS-PAGE and transferred onto PVDF membranes. Blocking was performed for 1 h with 5% non-fat milk/TBST buffer, followed by incubation overnight with primary antibodies (Rabbit polyclonal anti-LIN28A, CST, #3978, 1:1000; Mouse monoclonal anti-RPA194, Santa Cruz Biotechnology, sc-48385, 1:500; Rabbit monoclonal anti-GAPDH, CST, #5174,1:2000; Rabbit monoclonal anti-ACTIN, CST, #4970,1:2000) at 4 °C. The next day, the membranes were incubated with the appropriate secondary antibodies conjugated to HRP for 1 h at room temperature, and the bands were detected by ECL reagent and autoradiography.

## RNA extraction and qRT-PCR
Total RNA was extracted from mouse ESCs using the miRNeasy kit (217004, QIAGEN) according to the manufacturer's protocol, and 1 μg RNA was reversed transcribed to cDNA using the HiScript II Q RT Super Mix (R223-01, Vazyme). qPCR was performed with the SYBR-Green qPCR Master Mix (Bio-Rad) on a Bio-Rad PCR machine (CFX-96 Touch). Each gene was normalized to *Actin* or *Gapdh*. The primers used are listed in Supplementary Table 1.

## Immunofluorescence staining
Related ESCs were fixed with 4% PFA for 30 min, followed by permeabilizing with 0.5% Triton X-100/PBS for 20 min at RT. The cells were blocked in a blocking buffer (3% BSA, 2%donkey serum in PBS) for 10 min. Subsequently, the cells were stained with primary antibody (Rabbit polyclonal anti-LIN28A, CST, #3978, 1:200; Mouse monoclonal anti-FBL, Abcam, ab4566, 1:200; Mouse monoclonal anti-NPM, Sigma-Aldrich, B0556, 1:200; Rabbit monoclonal anti-NPM, Abcam, ab183340,1:100; Rabbit monoclonal anti-NANOG, Abcam, ab214549, 1:200) overnight at 4 °C. The next day, they were washed three times for 10 min with PBS, and the cells were stained with the appropriate Alexa Fluor conjugated secondary antibody for 1 h at 37 °C. Following washing three times with PBS, DAPI was used for nucleus staining. The cells were detected using the Zeiss LSM800 fluorescence microscope at a 63× oil objective. For high resolution microscopy imaging, LSM800 with the Airyscan module was used.

## OP-Puro labeling
To measure protein synthesis, mES cells were plated on gelatinized glass coverslips on MEF and recover overnight before additional treatment in LIF/2i or FGF2/Activin A medium respectively. The next day, cells was cultured in the medium containing CHX (50 μg/ml) for 30 min and labeling experiments were performed using Click-iT® Plus OPP Protein Synthesis Assay Kit (Life Technologies, C10456). After labeling, the cells were fixed with 4% paraformaldehyde for 15 min at room temperature and then permeabilized with 0.5% Triton X-100 for 15 min at room temperature according to the manufacturer's instructions. Nuclei were stained with DAPI for 2 min and the cell were observed under Zeiss LSM800 fluorescence microscope.

## Stimulated Emission Depletion Microscopy (STED) imaging
All cells were were fixed with 4% PFA for 30 min, followed by permeabilizing with 0.5% Triton X-100/PBS for 20 min at RT. The cells were blocked in a blocking buffer (3% BSA, 2%donkey serum in PBS) for 10 min. Subsequently, the cells were stained with primary antibody against FBL (Abcam, ab4566;1:200), NPM (Sigam, B0556; 1:200 and Abcam, ab183340,1:100), LIN28A (CST, 3978 S; 1:200) overnight at 4 °C. The next day, they were washed three times for 10 min with PBS, and the cells were stained with a secondary antibody (Anti-mouse IgG Alexa Fluor® 647 Conjugate, Jackson ImmunoResearch, 115-605-003,1:400; atto 488-goat anti-rabbit IgG, Sigma, 18772, 1:150) for 1 h at 37 °C. Following washing three times with PBS, DAPI was used for nucleus staining. STED images were acquired using Abberior Instruments with z-stack module. The *x*, *y*, and *z* axis resolution was 30 nm. STED Resol. was 5%. The images were analyzed using Fiji/ImageJ. In order to quantify nucleolus regularity, Boyce-Clark index was used[37]:

$$sbc = \sum_{i=1}^{n} |\left(\frac{ri}{\sum_{i=1}^{n}ri}\right)100 - \frac{100}{n}|$$

$r_i$ is the length from the vantage center to the boundary (semidiameter) of the nucleolus, n is the number of the semidiameter. FBL regularity data were up to the nearest integer in Fig. 7g.

## Fluorescence recovery after photobleaching (FRAP) analysis
Mouse E14 wild-type, knockout ESCs and mutant cells transduced with Lenti-FBL-mCherry lentivirus were cultured on MEF cells. FRAP experiments were performed on a ZEISS (Jena, Germany) LSM800 confocal laser scanning microscope equipped with a ZEISS Plan-APO 63x/NA1.46 oil immersion objective in a live cell imaging chamber. Circular regions of constant size were bleached with 100% laser power and monitored over time for fluorescence recovery. Bleaching was once every 5 s for a total of 5 min. Fluorescence intensity data were corrected for background fluorescence and normalized to initial intensity before bleaching and the FRAP curves were fitted using the GraphPad Prism v.8.2 software.

## Recombinant proteins expression and purification
LIN28A wild-type and mutant proteins were expressed from E.coli Condon Plus (DE3) cells (Agilent) and purified under native conditions unless otherwise noted. LIN28A-expressing constructs contain a TEV cleavage site between the N-terminal MBP tag and the target protein. E. coli was grown to OD600 of 0.6–0.8 and induced with 1 mM IPTG at 16°C overnight. Pelleted cells were resuspended in lysis buffer (50 mM Tris 7.0, 1 M NaCl, 5% glycerol, 1 mM DTT). After French Press, the

lysates were pelleted at 13201 ×g at 4 °C for 45 min. The supernatants were applied to Ni-NTA by gravity at about 1 mL/min at 4 °C. Fusion proteins were eluted by lysis buffer containing 300 mM Imidazole. After centrifugation, the protein elution was loaded to HiPrep™ 26/10 Desalting (Cytiva) equilibrated in PBS buffer. The eluted proteins were incubated with TEV protease at 4 °C overnight. Cleaved proteins were further purified with the NiNTA column, and remove the nucleic acids contamination by Capto S Column (Cytiva). The fractions were analyzed with SDS-PAGE, pooled, concentrated, flash frozen in liquid nitrogen, and stored at 80 °C.

### Protein labelling
The above LIN28A protein was labeled with AbFluor 488 using Lin-Kine™ 488 Labeling Kit (Abbkine), following the manufacturer's instructions. Briefly, recombinant proteins were incubated with the protein reaction buffer in 20 mM HEPES pH 8.3, 50 mM NaCl, 1 mM DTT, and rotated for 1 h at room temperature. Free dye was removed by the purification column.

### In vitro phase separation assay
In vitro phase separation assay was performed in a reaction buffer containing 50 mM HEPES, 50 mM NaCl, 1 mM DTT and 10% PEG. Protein concentration was determined using NanoDrop 3000. Enriched rRNA fragments from agarose gels with RNA Recovery Kit (ZymoResearch). Labeled LIN28A protein and RNA were mixed. The mixtures were loaded onto glass-bottom dishes (Cellvis), and imaging was performed using ZEISS (Jena, Germany) LSM800 Confocal microscope with 63× oil immersion objective.

### Electrophoretic mobility shift assay
*Pre-let-7g* RNA was labeled with Cy5 fluorescent dye. Reactions were performed with 20 nM labeled *pre-let-7g* probes incubated with increasing concentrations of protein in a buffer containing 10 mM HEPES (pH 8.0), 50 mM KCl, 1 mM EDTA, 1 mM DTT, 0.05% Triton X-100, 5% Glycerol, 0.01 mg/ml BSA, 40 U/mL RNase inhibitor (Beyotime, R0102). Reactions were incubated for 30 min and resolved on 5% native polyacrylamide gels.

### Reprogramming assays
One day in advance, 25,000 NHDF (or MEF) cells were plated per well in 12-well plates. The next day, cells were infected overnight with OCT4, SOX2, NANOG, LIN28A lentiviral factors at high multiplicity of infection. Two days later, cells were trypsinized and replated onto MEF feeder coated 6 or 12-well plates. Cells were then fed with ESC medium gradually until day 21(or 14) when plates were fixed. Reprogramming efficiency was evaluated by counting the number of iPSC colonies stained with alkaline phosphatase (AP) (Sigma-Aldrich).

### RNA-seq data analysis
All RNA-seq reads were trimmed using Trimmomatic software[56] (Version 0.36) with the following parameters "ILLUMINACLIP:TruSeq3-PE.fa:2:30:10 LEADING:3 TRAILING:3 SLIDINGWINDOW:4:15 MINLEN:36" and were further quality-filtered using FASTX Toolkit (http://hannonlab.cshl.edu/fastx_toolkit/) fastq_quality_trimmer command with the minimum quality score 20 and minimum percent of 80% bases that has a quality score larger than this cutoff value. The high-quality reads were aligned to the mm10 genome by HISAT2 (v2.1.0), a fast and sensitive spliced alignment program for mapping RNA-seq reads, with -dta parameter[57]. PCR duplicate reads were removed using Picard tools (https://broadinstitute.github.io/picard/) (v2.18.2). The expression levels of genes were calculated by StringTie[58] (Version v1.3.4d) with -e -B -G parameters using Release M18 (GRCm38.p6) gene annotations downloaded from GENCODE data portal. To obtain reliable and cross-sample comparable expression abundance estimation for each gene, reads mapped to mm10 were counted as TPM (Transcripts Per Million

reads) based on their genome locations. All statistical analyses and plots for Next Generation Sequencing (NGS) data were performed with R (v4.0.2)/Bioconductor (v3.10) software utilizing custom R scripts.

### Reporting summary
Further information on research design is available in the Nature Portfolio Reporting Summary linked to this article.

## Data availability
The data generated in this study are provided in the Supplementary Information/Source Data file. Source data are provided with this paper.

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

## Acknowledgements

We thank Dr. Xiong Ji and Dr. Peng Du (Peking University) for discussion and providing the Pol I degraded ESC lines. We thank Dr. Pengxu Qian (Zhejiang University) for providing the snoRNA knockout ESC lines. We thank Dr. Yafei Yin and Chong Tong (Zhejiang University) for giving a guide to the EMSA experiment. We thank Qin Han and Wei Yin from the core facility platform of Zhejiang University School of Medicine for their technical support. We thank Shichun Shao from the core facility platform of Liangzhu Laboratory for her technical support. J.Z. is supported by the National Key Research and Development Program of China (No.2018YFC1005002, No.2018YFA0107100, No.2018YFA0107103), the National Natural Science Foundation of China (No.31871453, No.91857116), the Zhejiang Natural Science Foundation of China (No.LR19C120001) and the Zhejiang Innovation Team Grant (2019R01004). H.Y. is supported by the National Natural Science Foundation of China (No.32100632) and the Zhejiang Natural Science Foundation of China (No.LQ21C120002).

## Author contributions

J.Z. and T.T. conceived the project. T.T. performed most of the experiments. T.T. and J.Z. discussed and wrote the manuscript. B.G. performed all recombinant proteins expression and purification. H.Y. performed the bioinformatics analysis. H.P. helped the immunofluorescence staining experiments. Z.S. provided LIN28A knockout cells. A.L., L.Z., H.L. helped with the reprogramming experiment. H.W. and G.D. gave constructive

advice on the project. Y.F. helped with the proteins expression and purification, and discussion the manuscript.

## Competing interests

The authors declare no competing interests.
