## [Peer Review File · Nature Communications]

Dynamic Nucleolar Phase Separation Influenced by Non-canonical Function of LIN28A Instructs Pluripotent Stem Cell Fate DecisionsEditorial Note: Parts of this Peer Review File have been redacted as indicated to remove third-party material where no permission to publish could be obtained.

REVIEWER COMMENTS

Reviewer #1 (Remarks to the Author):

In this study, Tan and colleagues investigated the role of LIN28A and nucleolar phase separation in pluripotent stem cells. The authors show that LIN28A undergoes phase separation in the nucleolus of pluripotent stem cells. They also demonstrate rRNA is essential for the phase separation of LIN28A and associated nucleolar proteins, such as FBL and NCLs. Furthermore, they show both LIN28A IDR and RBD domains are critical for the phase separation of LIN28A and associated nucleolar protein, and also find the key amino acids critical for the phase separation. Finally, using Lin28a KO cells and these Mut LIN28A, they suggest that the non-canonical phase separation property of LIN28A regulates nucleolar remodeling and cell fate decisions of pluripotent stem cells.

Major concerns

1, It is important to clarify the difference of the phase separation and other functions of LIN28A. Thus, it is required to demonstrate the only affecting phase separation but not other functions of LIN28A has resulted in the affects. More data need to be provided.

2, For the function study, the authors use the Human cells for iPS, but use Mouse cells for Primer-to-Naive transition. These two are different. Please use do iPS and Primer-to-Naive transition in both mouse and human cells.

2. The authors conclude that the nucleolar phase separation property, but not the canonical microRNA let-7 binding properties of LIN28A, that could mediate the naive-to-primed state conversion. It seems that a considerable portion of these mutant LIN28A protein is located in the cytoplasm, which may bind to RNA target. The authors need to provide data evidences that these mutant LIN28A lose its binding to microRNA let-7.

3. The authors show that phosphor-null LIN28A lose phase separation property and capacity to promote naïve-primed transition and reprogramming. They also propose that LIN28A is important for the phase stratification of nucleoli during reprogramming. Is phosphorylation of LIN28A important for the function of LIN28A in pluripotent stem cell fate? The authors needs to measure LIN28A phosphorylation in naïve-primed transition and reprogramming, and provide data of LIN28A phosphorylation function in naïve-primed transition and reprogramming.

4, What is the function of the phase separation of LIN28A in differentiation of pluripotent stem cells. The authors need to provide data of mouse and human cells.

5, Besides RNA, other proteins, ions or components are involved in the phase separation of LIN28A? In vitro and in vivo mouse and human data should be provided.

Minor concerns

1. There are some language errors in the manuscript, which need to be revised carefully.

Line 70-71

Linker and C-terminal regions of LIN28A, which includes S120, R192 and S200,...

Line 80-81

LIN28A protein undergoes ... and localization in...

Reviewer #2 (Remarks to the Author):

In this paper by Tan et al., the authors reported that LIN28A, an RNA binding protein could undergo nucleolar phase separation, which was linked to LIN28A's abilities to induce naïve to primed state transition in mouse embryonic stem cells. The authors pinpointed the domains and residues in LIN28A that assisted their phase separation. These domains also happened to be important for naïve to primed pluripotency transition. The paper is potentially interesting to the fields of phase separation, nucleolar biology, and stem cell differentiation. However, it suffers from technical deficiencies in phase separation characterizations. A lot of essential data is also descriptive/only showing representative images, lacking quantifications. The article seems to compile a number of experiments without clear explanation of their results. Therefore, my enthusiasm for the paper is dampened.

Major concerns:

1. Technical challenges in phase separation experiments:

a. Most of the phase separation experiments are done using an over-expressed LIN28A-GFP construct. How its behavior compares to endogenous LIN28A localization/ behavior is unknown. The STORM image showing endogenous LIN28A organization (Fig. 1i) looks different from overexpressed LIN28A-GFP.

b. 1,6-Hexandiol experiment is shown without any rationale. It is unclear what the authors mean by Hex treatment leads to diffusion of the condensates (clearly not the case judged by Fig. 1b). Plus, the concentration of Hex is too high (10% for 10 min). The nucleus should be dissolved by then. People usually use 1-3% for 10 min.

c. It is unclear how the authors performed FRAP experiments. Did the authors bleach half of the nucleolus, or the whole nucleolus? It is critical to explain, and to show the bleach images since to probe interior fluidity of the nucleolus, the authors should bleach half of it and analyze its recovery. If they bleached the whole, since there is not much LIN28A in the nucleoplasm, it is understandable that the recovery will be slow and not complete, because there is simply not much LIN28A surrounding the nucleolus to diffuse from. If this methodology is not established, any difference they see comparing WT and mutant LIN28A does not make much sense. It is also unclear FRAP in the cytoplasm is done. Are there cytoplasmic condensates of LIN28A too?

d. The authors found most of the domains of LIN28A are important for proper organization of nucleolus. However, they decided to say that only IDRs are important, and are the ones they follow up with using individual mutations. This is a bit puzzling and needs better explanation.

e. It is unclear if LIN28A leads to nucleolus formation, or is simply incorporated into the nucleolus. The authors seem to suggest that LIN28A and rRNA phase separation is sufficient to form the nucleolus (from the title of the paper and the in vitro data). But most likely LIN28A just incorporates into the existing nucleolus. If that's the case, then Figure 2 will not make much sense since depleting rRNA will only affect LIN28A phase separation indirectly by disrupting the nucleolus. The authors need to be less ambiguous about this important aspect of the paper.

2. A lot of essential data is also descriptive/only showing representative images, lacking quantifications. This is one of the main concerns of the paper. For example, the authors didn't use quantitative methods to show their data, but instead used descriptive and ambiguous ways of describing the nucleolar morphology (for example, using terms such as "floc", "diffused nucleoli", "fragmented nucleoli", "wreath structure"). These terms for a non-expert reader are hard to understand, and can potentially introduce biases when they quantify their images. Figures such as 2g, 3c, 4e, and 5h etc will benefit greatly from quantifications.

3. The article seems to compile a number of experiments without clear explanation of their results. The article as a whole has a lot of results that are not discussed very well, or seems out of place. For example, the cold-responsiveness of LIN28A phase separation seems out of place. I am also not sure what it means to have a slower FRAP recovery after hex treatment. The authors need to connect and discuss their data instead of simply presenting them.

Other points:

1. Lines 104-105: data in Fig. 2f doesn't support the claim that LIN28A expression level doesn't change: there is a decrease in LIN28A at 42C. GAPDH blot also seems to be cropped/manipulated.

2. Fig. 1i: separation of LIN28A from FBL doesn't automatically mean it colocalizes with other markers of GC and DFC. To demonstrate this, the authors need to show colocalization of LIN28A with GC and DFC.

3. Line 128: in vivo should be changed to in cell.

4. Line 199: the authors didn't include the in vitro data showing poor solubility and prone to aggregation.

5. Figure 5g, y-axis should be "relative irregularity of FBL"

Reviewer #1 (Remarks to the Author):

In this study, Tan and colleagues investigated the role of LIN28A and nucleolar phase separation in pluripotent stem cells. The authors show that LIN28A undergoes phase separation in the nucleolus of pluripotent stem cells. They also demonstrate rRNA is essential for the phase separation of LIN28A and associated nucleolar proteins, such as FBL and NCLs. Furthermore, they show both LIN28A IDR and RBD domains are critical for the phase separation of LIN28A and associated nucleolar protein, and also find the key amino acids critical for the phase separation. Finally, using Lin28a KO cells and these Mut LIN28A, they suggest that the non-canonical phase separation property of LIN28A regulates nucleolar remodeling and cell fate decisions of pluripotent stem cells.

We appreciate the reviewer's brief and comprehensive summary and thank for the helpful comment on our findings. Here we provide our full point-by-point responses below.

Major concerns

1. It is important to clarify the difference of the phase separation and other functions of LIN28A. Thus, it is required to demonstrate the only affecting phase separation but not other functions of LIN28A has resulted in the affects. More data need to be provided.

RESPONSE: We thank the reviewer for this constructive comment. LIN28A was originally discovered as a heterochronic gene regulating developmental timing in worms¹.

Conventionally, the role of LIN28A in cell fate decision is mainly attributed to its cytoplasmic functions in binding the precursors of microRNA let-7 and certain mRNA²⁻⁵. Mouse Lin28A has two folded RNA binding regions, the CSD and the CCHCx2, connected by a flexible linker of 14 amino acids, with extensions of ~30 residues at both the amino and carboxyl termini.

In this previous paper (PMID: 22078496), the authors showed that the CSD provides a larger contact and contributes to let-7 affinity, but the latter domain has additional effector functions, and neither the terminal nor linker regions outside of the folded domains (CSD and CCHCx2) are essential for blocking let-7 *in vivo*⁶.

In mice and human, the let-7 family miRNAs comprise 12 members, the mature miRNA sequence of which is highly conserved between the different genes. LIN28A binds precursor forms of let-7 miRNAs and can inhibit both pri-let7 processing by Drosha and pre-let7 processing by Dicer⁶. In order to address the “other functions” such as RNA binding raised by the reviewer, we first focused our analysis on the pre-let-7g, as previous studies had indicated its importance in direct association with LIN28A.

Mouse recombinant LIN28A was prepared as described in the main text of the revised manuscript. Using electrophoretic mobility shift assay (EMSA), we analyzed a series of single amino acid mutants of LIN28A to distinguish whether these mutants affected the phase separation function or the let-7 binding function of LIN28A. Both the S120 (Located in the linker disordered region) and R192 (located in the C terminal disordered region), S200 (Located in C-terminal disordered regions) mutations did not affect the LIN28A binding to pre-let-7g, but the mutation of F47 (Located at the CSD domain) abolished LIN28A binding to pre-let-7 (**Response Figure 1a,1c**). F47 has already been identified as a single amino acid residues required for binding to pre-let-7g, so it is used as a positive control for pre-let-7 binding. In agreement with the EMSA result, qPCR analysis also showed that S120, S200 and R192 can be mutated without affecting the expression of mature let-7g(**Response Figure 1b**).

As a RNA-binding protein, LIN28A can also bind mRNA (such as *Ndufb10* RNA, an OxPhos Gene mRNA) and regulates stem cell metabolism^{5,7,8}. Using electrophoretic mobility shift assay (EMSA), we found that WT LIN28A and LIN28A IDR point mutations had similar capacity for binding to *Ndufb10* mRNA (**Response Figure 1d**).

Together the above data demonstrated that the IDR mutations did not affect LIN28A's function in RNA binding and regulation, at least for the microRNA let-7g and the mRNA we tested.

In terms of phase separation, the localization and morphology of these mutant LIN28A protein in the nucleolus were changed compared with WT LIN28A, but the localization and morphology in the cytoplasm was unchanged (**Response Figure 1e**). FRAP analysis also showed that these mutant LIN28A protein had lower fluidity in the nucleolus, and so as the FBL protein (**Response Figure 1f,1g**). To ultimately confirm the nucleolar disruptive role of LIN28A mutants was resulted from their weakened phase separation property, we generated the rescuing LIN28A mutants by fusing the exogenous IDR of FUS (S120A-FUS IDR, S200A-FUS IDR, R192G-FUS IDR), which is known to drive phase separation. Notably, the fused IDRs rescued the morphology and phase separation capability of the LIN28A mutants (**Response Figure 1e,1f**). Meanwhile, the three IDR fusions completely rescued the impaired fluidity of FBL caused by the LIN28A mutants (**Response Figure 1g**).

Together, these results demonstrated that the key amino acids at the IDR regions were essential for LIN28A phase separation function in the nucleolus, as well as for maintaining normal nucleolar integrity. On the other hand, these IDR mutations did not affect LIN28A's canonical functions such as RNA binding and regulation, at least for the microRNA let-7g and the mRNA we tested. We can not completely rule out there are other RNAs that can be affected by these IDR mutations, nor rule out other functions beyond RNA binding. But what is certain is that these mutations do affect LIN28A nucleolar phase separation and nucleolar integrity.

Response Figure 1

Response Figure 1

- Constructs used to investigate the function of mutations of mouse LIN28A. (also new **Extended Data Fig.8a in the revised manuscript**)
- qRT-PCR showing the *let-7g* expression in WT, *Lin28a* KO, and LIN28A mutant ESCs. $n = 3$ biologically independent experiments, error bar: standard error of the mean, * $p < 0.05$, ** $p < 0.01$, *** $p < 0.001$, Two-way ANOVA. (also new **Extended Data Fig.8b in the revised manuscript**)
- EMSA with pre-*let-7g* as the probe, mixed with increasing concentrations (0.02, 0.1, 0.2, 0.5, and $1\mu\text{M}$) of mouse LIN28A variants. (also new **Extended Data Fig.8c in the revised manuscript**)
- EMSA with *Ndufb10* 3' RNA oligo as the probe, mixed with increasing concentrations (0.2, 0.5, 2, 5, and $10\mu\text{M}$) of mouse LIN28A variants.
- Representative confocal microscopy Airyscan images of the morphology and nucleolar localization of LIN28A and FBL in living WT, *Lin28a* KO, and *Lin28a* KO cells transduced with Mut-LIN28A, S120A LIN28A, S200A LIN28A, R192G LIN28A, and FUS protein's IDR fused LIN28A variants cultured in LIF/Serum medium. Scale bar, $5\mu\text{m}$. (also **Fig.4f in the revised manuscript**)
- FRAP analysis showing fluorescence recovery of WT-LIN28A, S120A, S200A, R192G, and IDR fusion variants after photobleaching in the nucleus ($n = 3$) in biologically independent experiments. Data are presented as mean values \pm SEM. SEM: standard error of the mean. (also **Fig.4d in the revised manuscript**)
- FRAP analysis showing FBL fluorescence recovery after photobleaching in WT, *Lin28a* KO, and *Lin28a* KO cells transduced with S120A LIN28A, S200A LIN28A, R192G LIN28A, and FUS protein's IDR-fused LIN28A variants ($n = 3$) in biologically independent experiments. Data are presented as mean values \pm SEM. SEM: standard error of the mean. (also **Fig.4i in the revised manuscript**)

2. For the function study, the authors use the Human cells for iPS, but use Mouse cells for Primer-to-Naive transition. These two are different. Please use do iPS and Primer-to-Naive transition in both mouse and human cells.

RESPONSE: We thank the reviewer for this comment. The mouse and human naive vs primed regulation mechanisms are different. LIN28A itself does not play a role in the human naive-to-primed transition, as discussed below, therefore, it is not necessary to study its variants' function in the context of naive-to-primed transition. However, according to the reviewer's comment about mouse and human iPSCs, now we have further performed reprogramming of mouse embryonic fibroblast cells into mouse iPSCs in the revised manuscript (in addition to the reprogramming of human fibroblast cells into human iPSCs in the original manuscript). LIN28A IDR point mutations led to reduced reprogramming efficiency in this new mouse reprogramming experiment.

Moreover, to further validate the inhibitory role of LIN28A mutants on reprogramming resulted from their impaired phase-separation capacity, we generated the rescue LIN28A variants by fusing the IDR of FUS, which is known to drive phase separation. FUS IDR fusion rescued the the reduced reprogramming efficiency (**Response Figure 2a,2b**).

We further characterized the mouse iPSCs. Stimulated Emission Depletion Microscopy (STED) imaging revealed significant differences in the morphology and stratification of the nucleoli between MEF cells and mouse iPSCs. In the MEF cells, NPM was in irregular shape, whereas in the iPSCs, NPM showed the round 'lotus root' structure and was colocalized with LIN28A(**Response Figure 2c**). We further quantified the regularity of the granular component (GC) using Boyce-Clark semidiameter index which was originally used to assess the 'compactness' of space layouts⁹. GC showed higher degree of regularity in iPSCs compared with MEF(**Response Figure 2e**). Besides, the STED imaging showed the DFC component FBL tended to show the 'ring' structure indicating more developed DFC units and was embedded and immersed within the granular component NPM in the iPS cells(**Response Figure 2d,2f**). This demonstrated the nucleoli were clearly stratified(**Response Figure 2g**). OP-Puromycin staining indicated that iPSCs possessed higher protein synthesis rate compared with MEF cells, suggesting the more clearly stratified nucleoli in iPSCs were functionally more developed (**Response Figure 2h**).

Taken together, these results demonstrated that in the context of mouse, the IDR region of LIN28A that regulated phase separation played an important role in reprogramming and mouse iPSCs.

Finally, the situation and the regulation mechanisms for mouse and human naive and primed states are different, LIN28A does not have a reported role in human naive-to-primed state transition and is not a marker of naive or primed state¹⁰⁻¹³. Our qPCR analysis was also consistent with this conclusion: unlike other naive (KLF4, KLF17, DPPA3) and primed (CD24) genes that were down-regulated or up-regulated respectively during human naive-to-primed conversion, LIN28A did not change its expression level during this process (**Response Figure 2i,2j**).

Response Figure 2

Response Figure 2

- Reprogramming efficiency of MEF cells transduced with OCT4, SOX2, NANOG, and LIN28A variants. (also new Fig.6a in the revised manuscript)
- Number of iPS colonies 14 days after OSNA transduction of MEF cells. $n = 3$ biologically independent experiments, error bar: standard error of the mean, $*p < 0.05$, $**p < 0.01$, $***p < 0.001$; One-way ANOVA. (also new Fig.6b in the revised manuscript)
- Representative STED immunofluorescence images of LIN28A and NPM in MEF or iPSC cells. (also new Fig.6c in the revised manuscript)
- Representative STED immunofluorescence images of NPM and FBL in MEF or iPSC cells. (also new Fig.6d in the revised manuscript)

- e. Statistical analysis of the GC irregularity in MEF and iPSCs using Boyce-Clark semidiameter index. The larger the number, the more irregular the FBL. n=20; *p < 0.05, **p < 0.01, ***p < 0.001, unpaired two-tailed student's t-test. **(also new Fig.6e in the revised manuscript)**
- f. Graph showing the number of typical DFC(FBL) morphology in MEF or iPSCs. n=20 nucleoli, number of nucleoli. **(also new Fig.6f in the revised manuscript)**
- g. A cartoon diagram showing morphological changes of NPM and FBL in MEF or iPSCs. **(also new Fig.6g in the revised manuscript)**
- h. Immunofluorescence imaging showing OP-puromycin-labeled MEF and iPSCs. Scale bar, 20µm. **(also new Fig.6h in the revised manuscript)**
- i. qRT-PCR showing the naïve pluripotent marker gene expression in human naïve state cells and primed state cells. n = 3 biologically independent experiments, error bar: standard error of the mean, *p < 0.05, **p < 0.01, ***p < 0.001, Two-way ANOVA.
- j. qRT-PCR showing the primed pluripotent marker gene and *LIN28A* gene expression in human naïve state cells and primed state cells. n = 3 biologically independent experiments, error bar: standard error of the mean, *p < 0.05, **p < 0.01, ***p < 0.001, Two-way ANOVA.

3. The authors conclude that the nucleolar phase separation property, but not the canonical microRNA let-7 binding properties of LIN28A, that could mediate the naive-to-primed state conversion. It seems that a considerable portion of these mutant LIN28A protein is located in the cytoplasm, which may bind to RNA target. The authors need to provide data evidences that these mutant LIN28A lose its binding to microRNA let-7.

RESPONSE: We thank the reviewer for this good and constructive comment. In the process of our experiments, WT Lin28A and the three LIN28A IDR mutants had similar localization and fluidity in the cytoplasm. Using electrophoretic mobility shift assay (EMSA), we found that WT Lin28A and LIN28A IDR point mutations had similar capacity for binding to microRNA let-7(pre-let-7g)(**Response Figure 3a**). qPCR analysis also showed that S120, S200 and R192 can be mutated without affecting the expression of mature let-7g (**Response Fig. 3b**). These data illustrated that the IDR mutations did not affect LIN28A binding to microRNA let-7, nor let-7 expression level.

On the other hand, to prove that the IDR mutations can cause disruption of nucleolar phase separation through the phase separation function of LIN28A, we generated the rescuing LIN28A mutants by fusing the exogenous IDR of FUS (S120A-FUS IDR, S200A-FUS IDR, R192G-FUS IDR), which is known to drive phase separation¹⁴⁻¹⁶. Notably, the fused IDRs rescued the morphology and phase separation capability of LIN28A itself and other nucleolar protein FBL in the LIN28A mutants(**Response Figure 3c, 3d, 3e**).

Finally, we showed by NANOG immunostaining(**Response Figure 3f,3g**) and qPCR analysis(**Response Figure 3h**) that the IDR mutants led to reduced naive-to-primed conversion efficiency, and it was through the phase separation function of LIN28A because the FUS IDR fusion could completely rescue the defects caused by the IDR mutants.

Response Figure 3

Response Figure 3

- EMSA with *pre-let-7g* as the probe, mixed with increasing concentrations (0.02, 0.1, 0.2, 0.5, and 1 μ M) of mouse LIN28A variants. (also new Extended Data Fig.8c in the revised manuscript)
- qRT-PCR showing the *let-7g* expression in WT, Lin28a KO, and LIN28A mutations ESCs. n = 3 biologically independent experiments, error bar: standard error of the mean, *p < 0.05, **p < 0.01, ***p < 0.001, Two-way ANOVA. (also new Extended Data Fig.8b in the revised manuscript)
- FRAP analysis showing fluorescence recovery of WT-LIN28A, S120A, S200A, R192G, and FUS-IDR fused LIN28A variants after photobleaching in the nucleus (n = 3) in biologically independent experiments. Data are presented as mean values \pm SEM. SEM: standard error of the mean. (also Fig.4d in the revised manuscript)
- FRAP analysis showing FBL fluorescence recovery after photobleaching in WT, Lin28a KO, and Lin28a KO cells transduced with S120A LIN28A, S200A LIN28A, R192G LIN28A, and FUS-IDR-fused LIN28A variants (n = 3) in biologically independent experiments. Data are presented as mean values \pm SEM. SEM: standard error of the mean. (also Fig.4i in the revised manuscript)
- Representative confocal microscopy Airyscan images of the morphology and nucleolar localization of LIN28A and FBL in living WT, Lin28a KO, and Lin28a KO cells transduced with Mut-LIN28A, S120A LIN28A, S200A LIN28A, R192G LIN28A, and FUS-IDR-fused LIN28A variants cultured in LIF/Serum medium. Scale bar, 5 μ m. (also Fig.4f in the revised manuscript)
- NANOG immunostaining of WT, Lin28a KO, and Lin28a KO ESCs transduced with full length WT LIN28A, single mutation variants, or IDR-fused variants, converted from the naïve state to the primed state. Scale bar, 200 μ m. (also Fig.5c in the revised manuscript)
- Statistical analysis of NANOG protein fluorescence intensity of the above cells in (f). n = 64, error bar: standard error of the mean, *p < 0.05, **p < 0.01, ***p < 0.001, Two-way ANOVA. (also Fig.5d in the revised manuscript)
- qRT-PCR showing the naïve and primed pluripotent marker gene expression in WT, Lin28a KO, and Lin28a KO ESCs transduced with full length WT LIN28A or the indicated LIN28A variants, cultured in the naïve and primed state conditions. n = 3 biologically independent experiments, error bar: standard error of the mean, *p < 0.05, **p < 0.01, ***p < 0.001, Two-way ANOVA. (also Extended Data Fig.7c in the revised manuscript)

4. The authors show that phosphor-null LIN28A lose phase separation property and capacity to promote naïve-primed transition and reprogramming. They also propose that LIN28A is important for the phase stratification of nucleoli during reprogramming. Is phosphorylation of LIN28A important for the function of LIN28A in pluripotent stem cell fate? The authors needs to measure LIN28A phosphorylation in naïve-primed transition and reprogramming, and provide data of LIN28A phosphorylation function in naïve-primed transition and reprogramming.

RESPONSE: We thank the reviewer for this helpful suggestion. In this previous paper (PMID: 27992407)(**Response Figure 4a**), the authors have demonstrated that LIN28A S200 phosphorylation contributes to the regulation of reprogramming or pluripotency state transition¹⁷.

First, it showed that the phosphor-null (S200A) and phospho-mimetic (S200D) LIN28A led to approximately 50% decreased or increased reprogramming efficiency(**Response Figure 4b**), respectively, indicating that S200 phosphorylation had an important role in the induction of pluripotency.

They then explored the regulation of LIN28A levels in mESCs cultured in serum/LIF versus 2i/LIF naive state conditions. Total LIN28A protein and phosphorylated LIN28A levels were reduced in the 2i/LIF culture(**Response Figure 4c**). In the serum/LIF condition (or a metastable state close to the primed state), due to the presence of the ERK signaling, the S200 of LIN28A is phosphorylated which leads to higher stability of the LIN28A protein(**Response Figure 4d,4e,4g**). So in the primed state or the serum/LIF condition, the total LIN28A level, as well as the phosphorylated LIN28A level are higher compared with the naive state. On the other hand, in the naive condition, because the ERK signaling is inhibited, S200 of LIN28A is not phosphorylated(**Response Figure 4f**), and thus LIN28A protein stability is lower, and the protein level is also lower. This is how the phosphorylation of S200 of LIN28A regulates the naive/primed pluripotent states.

They then performed clonogenic assays upon transfer from 2i/LIF to serum/LIF and assessed the alkaline phosphatase (AP) staining pattern of colonies emerging in the serum/LIF culture, which is characterized by a mix of compact, uniformly AP-positive, naïve-like (“solid”) colonies and larger, heterogeneously AP-stained, more primed (“mixed”) colonies. The S200D phospho-mimetic mutant showed a reproducibly higher number of solid colonies relative to wild-type LIN28A(**Response Figure 4h**), demonstrating that the higher LIN28A protein level mediated by S200 phosphorylation enhances LIN28A’s function in promoting the transition from naïve to primed pluripotency.

Together, these reprogramming and mESC data demonstrate that LIN28A phosphorylation contributes to reprogramming and the regulation of pluripotency state transition.

Response Figure 4

[REDACTED]

Response Figure 4

- a. The referenced article. Response Figure 4 is from this article.
- b. TRA-1-60 staining of iPSCs from a reprogramming experiment using OSN and empty vector (EV), wild-type LIN28A (WT), phospho-null (S200A) LIN28A, or phospho-mimetic (S200D) LIN28A (day 21 of reprogramming). Western blot analysis of endogenous LIN28A in v6.5 mESCs cultured in serum/LIF or 2i/LIF.
- c. Western blot analysis of endogenous LIN28A in v6.5 mESCs cultured in serum/LIF or 2i/LIF conditions.
- d. Western blot analysis of transgenic wild-type (WT), phosphomimetic (S200D), or phospho-null (S200A) FLAG-LIN28A added back in LIN28A/B KO mESCs.
- e. Cycloheximide chase of transgenic FLAG-LIN28A variants in HeLa (Flp-In) cells. CHX= cycloheximide (100 µg/ml).
- f. Western blot (left) analysis of endogenous LIN28A in v6.5 mESCs after a four-hour dropout of PD0325901 (PD) or CHIR99021 (CH).
- g. Western blot analysis of LIN28A (S200) phosphorylation in PA1 cells after 30min stimulation with serum (10%), fibroblast growth factor (FGF) (100 ng/ul), or epidermal growth factor (EGF) (100 ng/ul). Cells were serum-starved for 16–20 h prior to stimulation.
- h. Alkaline Phosphatase (AP) analysis of mESCs from panel (f) grown at clonal density upon transfer from 2i/LIF to serum/LIF. Representative images of colonies with solid and mixed staining patterns are shown on the left. Scale bar = 100 µm. Quantification of the number of solid colonies is shown on the right. n=4 independent experiments. Error bars represent s.e.m. **P<0.01 (two-tailed Student's t-test).

5. What is the function of the phase separation of LIN28A in differentiation of pluripotent stem cells. The authors need to provide data of mouse and human cells.

RESPONSE: We thank the reviewer for this good suggestion. The undifferentiated state of mouse ES cells is maintained in the presence of leukemia inhibitory factor(LIF) in the culture medium¹⁸⁻²⁰. The spontaneous differentiation of ES cells can be triggered by withdrawal of LIF from the medium^{21,22}.

In our experiments, the spontaneous differentiation was induced by the withdrawal of LIF 12 h after plating E14 ESCs. Cell pellets were collected at 3, 5, and 7 days after LIF withdrawal and induction of differentiation, and then qPCR analysis showed that the expression of *Lin28a* decreased sharply after LIF withdrawal in 3 days (**Response Figure 5a**). The IDR mutated LIN28A variants can slightly delay the differentiation of pluripotent stem cells (**Response Figure 5b**). The possible reason is that these mutants had the tendency to stay in a state closer to the naive state, and had slower kinetics to exit the pluripotency as illustrated in the Response Figure 3g, 3f, 3h above. As LIN28A does not play a role in human naive pluripotency exit, as illustrated in the Response Figure 2i, 2j, it is not necessary to examine the human situation, and we use the mouse ESC exit from pluripotency and induction of differentiation here to address the question.

Besides, it is well-established that LIN28A is a pluripotency factor. LIN28A was abundantly expressed in undifferentiated ESCs, embryonal carcinoma cells and early embryonic tissue, but declines in expression and becomes tissue restricted. Because LIN28A is decreased during differentiation, we expect that it does not have a role in regulating differentiation genes. Our qPCR analysis also showed that the IDR mutant LIN28A did not affect the expression kinetics of genes in the three germ layers(**Response Figure 5c,5d,5e**).

Response Figure 5

Response Figure 5

- a. qRT-PCR showing WT *Lin28a* gene expression after LIF withdrawal. n = 3 biologically independent experiments, error bar: standard error of the mean. **(also Extended Data Fig.10a in the revised manuscript)**
- b. qRT-PCR showing pluripotency factor *Nanog* gene expression in WT, Lin28a KO, and Lin28a KO cells transduced with FL-LIN28A, S120A-LIN28A, S200A-LIN28A, R192G-LIN28A, and FUS protein's IDR fused LIN28A variants after LIF withdrawal. n = 3 biologically independent experiments, error bar: standard error of the mean, *p < 0.05, **p < 0.01, ***p < 0.001, One-way ANOVA. **(also Extended Data Fig.10b in the revised manuscript)**
- c. qRT-PCR showing endoderm lineage gene *Gata6* expression in WT, Lin28a KO, and Lin28a KO cells transduced with FL-LIN28A, S120A-LIN28A, S200A-LIN28A, R192G-LIN28A, and FUS protein's IDR fused LIN28A variants after LIF withdrawal. n = 3 biologically independent experiments, error bar: standard error of the mean, *p < 0.05, **p < 0.01, ***p < 0.001, One-way ANOVA. **(also Extended Data Fig.10c in the revised manuscript)**
- d. qRT-PCR showing mesoderm lineage gene *T* expression in WT, Lin28a KO, and Lin28a KO cells transduced with FL-LIN28A, S120A-LIN28A, S200A-LIN28A, R192G-LIN28A, and FUS protein's IDR fused LIN28A variants after LIF withdrawal. n = 3 biologically independent experiments, error bar: standard error of the mean, *p < 0.05, **p < 0.01, ***p < 0.001, One-way ANOVA. **(also Extended Data Fig.10d in the revised manuscript)**
- e. qRT-PCR showing ectoderm lineage gene *Pax6* expression in WT, Lin28a KO, and Lin28a KO cells transduced with FL-LIN28A, S120A-LIN28A, S200A-LIN28A, R192G-LIN28A, and FUS protein's IDR fused LIN28A variants after LIF withdrawal. n = 3 biologically independent experiments, error bar: standard error of the mean, *p < 0.05, **p < 0.01, ***p < 0.001, One-way ANOVA. **(also Extended Data Fig.10e in the revised manuscript)**

6. Besides RNA, other proteins, ions or components are involved in the phase separation of LIN28A? In vitro and in vivo mouse and human data should be provided.

RESPONSE: We thank the reviewer for bringing our attention on this issue. The LIN28A protein and RNA were mixed at the indicated concentrations. Then in vitro phase separation assay was performed in a reaction buffer containing 50mM HEPES, 50mM NaCl, 1mM DTT and 10% PEG.

It's worth noting that **low salt promotes LIN28A phase separation in vitro**. At first, we tried to observe phase separation in a reaction buffer containing 50mM HEPES, 150mM NaCl (physiological salt conditions), 1mM DTT and 10% PEG, the solution of LIN28A proteins remained clear at room temperature (by visual inspection), and when examined by light microscopy, only the irregular aggregation were observed (**Response Figure 6a, 6b**). However, when the NaCl concentration was diluted to 50mM, solutions of LIN28A proteins became opalescent (by visual inspection), and round structures were observed by light microscopy (**Response Figure 6a,6b**).

The temperature plays a role in phase separation. LIN28A was localized in both the nucleus and cytoplasm at 37°C (**Response Figure 6c**). When cells were exposed to cold shock (25°C), LIN28A tended to be reduced in the nucleus (**Response Figure 6c**). When cells were exposed to heat shock (42°C) for 15min, LIN28A tended to become more compact compared with that at 37°C (**Response Figure 6c**). Next, we investigated the difference in fluidity of the LIN28A phase-separated condensate at 37°C, 25°C and 42°C. Fluorescence recovery after photobleaching (FRAP) analysis of LIN28A-eGFP was performed at the indicated temperatures. After bleaching for 300 seconds, fluorescence signals of LIN28A in the nucleus had 20-30% of recovery in all the three conditions, with no significant difference in the degree of LIN28A recovery (**Response Figure 6d**). In contrast, the fluidity of LIN28A in the cytoplasm had above 40% recovery and the 42°C condition had over 60% recovery (**Response Figure 6d**). Statistical analysis showed that cold shock decreased LIN28A loci numbers in the nucleus and the nucleus : cytoplasm intensity ratio, whereas heat shock increased both (**Response Figure 6e,6f**). We also observed more dispersed areas of LIN28A in cold-shocked nuclei and more compact areas of LIN28A in heat-shocked nuclei (**Response Figure 6g**). These results suggested that cold shock promoted LIN28A outflow from the nucleus and the condensate became more diffused, while heat shock promoted its inflow into the nucleus and the condensate became more compact.

The **intrinsically disordered regions (IDRs)** obviously play a role. A hallmark for LLPS assemblies for many proteins, is the presence of IDRs. These regions are enriched in repetitive sequences of a few amino acids, usually resulting in characteristic domains of low complexity²³. Besides RNA, the N/C-terminal and Linker IDRs promoted the establishment of the LIN28A protein phase separated condensate both in vitro (**Response Figure 6i,6j**) and in vivo (**Response Figure 6h**). The LIN28A truncation experiment and IDR mutation experiment demonstrated that the IDR regions were essential for LIN28A phase separation.

Regarding other proteins, we mainly investigated them in vivo, such as FBL and NCL, but not in vitro, because the in vitro reconstituted recombinant proteins can't actually reflect the real situation in vivo.

Regarding human and mouse LIN28A recombinant proteins, there are 97% similar in their protein sequence (**Response Figure 6k**), and mouse LIN28A protein is frequently used to indicate functions of LIN28A of both mouse and human⁶.

Response Figure 6

Response Figure 6

- a. Representative images of LIN28A recombinant protein forms irregular aggregation or droplets in the presence of total RNA (extracted from ES cells) in vitro. The in vitro phase separation assay was performed with 750 μ M LIN28A protein in reaction buffer containing 50 mM HEPES pH 7.5, 150 mM NaCl(high salt) or 50mM NaCl(low salt), 1 mM DTT, and 10% PEG-8000 (Sigma). Scale bar, 10 μ m. **(also new Extended Data Fig.1c in the revised manuscript)**
- b. Representative images of human LIN28A recombinant protein forms irregular aggregation or droplets in the presence of total RNA (extracted from ES cells) in vitro. The in vitro phase separation assay was performed with 150 μ M LIN28A protein in reaction buffer containing 50 mM HEPES pH 7.5, 150 mM NaCl(high salt) or 50mM NaCl(low salt), 1 mM DTT, and 10% PEG-8000 (Sigma). Scale bar, 10 μ m. **(also new Extended Data Fig.1d in the revised manuscript)**
- c. Confocal microscopy Airyscan imaging of the live eGFP-LIN28A cells at 37°C, 25°C and 42°C. Scale bar, 5 μ m.
- d. FRAP analysis showing temperature shock impacted fluorescence recovery after photobleaching of eGFP-LIN28A; n= 3 biologically independent experiments. Data are presented as mean values \pm SEM. SEM: standard error of the mean.
- e. Statistical analysis of LIN28A protein loci numbers in the nucleus at three different temperatures .37°C, 42°C and 25°C; n=40; *p < 0.05, **p <0.01, ***p < 0.001, One-way ANOVA.
- f. Statistical analysis of LIN28A nucleus/cytoplasm fluorescence intensity ratio at three different temperatures. 37°C, 42°C and 25°C; n=25; *p < 0.05, **p <0.01, ***p < 0.001, One-way ANOVA.
- g. Statistical analysis of LIN28A loci area in the nucleus at three different temperatures. 37°C, 42°C and 25°C; n=21; *p < 0.05, **p <0.01, ***p < 0.001, One-way ANOVA.
- h. FRAP analysis showing WT-LIN28A, S120A, S200A, R192G, and IDR fusions recovery after photobleaching in the nucleus (n = 3) in biologically independent experiments. Data are presented as mean values \pm SEM. SEM: standard error of the mean. **(also Fig.4d in the revised manuscript)**
- i. LLPS of purified recombinant LIN28A protein in 50 mM NaCl and 100 ng/ μ L total RNA. Scale bar, 10 μ m. **(also Fig.3g in the revised manuscript)**
- j. Summary of LLPS of purified recombinant LIN28A protein under indicated conditions, in the presence of 50 mM NaCl in vitro. **(also Fig.3h in the revised manuscript)**
- k. Mouse and human LIN28A proteins sequence alignment.

Minor concerns

1. There are some language errors in the manuscript, which need to be revised carefully.

Line 70-71

Linker and C-terminal regions of LIN28A, which includes S120, R192 and S200,...

Line 80-81

LIN28A protein undergoes ... and localization in...

RESPONSE: We thank the reviewer for bringing our attention on this issue. According to the reviewer's suggestion, we have corrected these language errors in our revised manuscript.

Besides, we decided to remove the temperature-related figures and text. We have deleted this statement (Line 80-81, LIN28A protein undergoes ... and localization in...).

The corrections are shown below:

Line 70-71 from "Linker and C-terminal regions of LIN28A, which includes S120, R192 and S200,..." to "**Linker and C-terminal regions of LIN28A, which include S120, R192 and S200,...**"

Reviewer #2 (Remarks to the Author):

In this paper by Tan et al., the authors reported that LIN28A, an RNA binding protein could undergo nucleolar phase separation, which was linked to LIN28A's abilities to induce naïve to primed state transition in mouse embryonic stem cells. The authors pinpointed the domains and residues in LIN28A that assisted their phase separation. These domains also happened to be important for naïve to primed pluripotency transition. The paper is potentially interesting to the fields of phase separation, nucleolar biology, and stem cell differentiation. However, it suffers from technical deficiencies in phase separation characterizations. A lot of essential data is also descriptive/only showing representative images, lacking quantifications. The article seems to compile a number of experiments without clear explanation of their results. Therefore, my enthusiasm for the paper is dampened.

We thank the reviewer for the positive comments, as quoted here “The paper is potentially interesting to the fields of phase separation, nucleolar biology, and stem cell differentiation.” We appreciate the critics that a few key questions need to be addressed to make these findings more convincing. We provide our full point-by-point responses below.

Major concerns:

1. Technical challenges in phase separation experiments:

a. Most of the phase separation experiments are done using an over-expressed LIN28A-GFP construct. How its behavior compares to endogenous LIN28A localization/behavior is unknown. The STORM image showing endogenous LIN28A organization (Fig. 1i) looks different from overexpressed LIN28A-GFP.

RESPONSE: We thank the reviewer for this constructive suggestion. Following this guidance, we have generated the eGFP-LIN28A knock-in mESCs line (**Response Figure 7a,7b**). LIN28A had the similar localization in the nucleolus and cytoplasm in the wild-type ES cells (by immunofluorescence staining), eGFP-LIN28A-overexpressed (live imaging), and eGFP-LIN28A knock-in mESCs line (live imaging) (**Response Figure 7c**).

Next, we tested the fluidity of LIN28A protein condensates both in the nucleolus and cytoplasm, FRAP experiments indicated that overexpressed LIN28A exhibited similar fluidity with the knock-in LIN28A (**Response Figure 7d**).

Due to the differences in image processing, the STORM image looks like a cartoon (we removed in our revised manuscript). In fact, LIN28A immunostaining in the wild-type, overexpressed eGFP-LIN28A and eGFP knock-in LIN28A live imaging all showed that LIN28A in the nucleolus formed a round shell with holes, and FBL was embedded in the holes. LIN28A was diffusely distributed in the cytoplasm (**Response Figure 7c**).

Response Figure 7

Response Figure 7

- Schematic representation of the eGFP-LIN28A knock-in E14 mESC line generated. **(also new Fig. 1a in the revised manuscript)**
- PCR experiments showing that the knock-in mESCs genome contains the inserted has an extra genome of about 700 base fragments compared with wild-type (WT) mES cells. This result indicates that the LIN28A-eGFP gene was successfully knocked-in (homozygotes). **(also new Extended Data Fig.1a in the revised manuscript)**
- Confocal microscopy Airyscan images of the morphology and nucleolar localization of LIN28A and FBL in immunostained wild-type E14 mESCs, live-imaging overexpressed LIN28A-eGFP overexpressing E14 mESCs, and live imaging LIN28A-eGFP knock-in E14 mESCs. Scale bar, 5µm. **(also new Fig. 1b in the revised manuscript)**
- FRAP analysis and images showing overexpressed eGFP-LIN28A and knock-in eGFP-LIN28A recovery after photobleaching in the nucleus ($n = 3$) in biologically independent experiments. Data are presented as mean values \pm SEM. SEM: standard error of the mean. Scale bar, 5µm. **(also new Extended Data Fig.1b in the revised manuscript)**

b. 1,6-Hexandiol experiment is shown without any rationale. It is unclear what the authors mean by Hex treatment leads to diffusion of the condensates (clearly not the case judged by Fig. 1b). Plus, the concentration of Hex is too high (10% for 10 min). The nucleus should be dissolved by then. People usually use 1-3% for 10 min.

RESPONSE: We thank the reviewer for this helpful suggestion. 1,6-hexanediol (HEX) is used as a LLPS inhibitor in phase separation studies²³, and we used it to indicate that LIN28A forms phase separated condensate. According to the reviewer's suggestion, we have updated and reanalyzed the experiment in the eGFP knock-in ESCs. ESCs were treated with 1% HEX for 10 minutes and we observed that LIN28A was diffused in the nucleus (**Response Figure 8a, 8d**). We also performed FRAP experiments and found that LIN28A in the nucleolus exhibited slower recovery compared to that in the cytoplasm (**Response Figure 8b, 8c**), and the nucleolar LIN28A condensate was more sensitive to the HEX treatment, suggesting LIN28A forms more less fluidy condensate in the nucleolus than in the cytoplasm.

Response Figure 8

Response Figure 8

- a. Confocal microscopy Airyscan imaging of the living knock-in eGFP-LIN28A mESCs with 1%HEX treatment. Scale bar, 10 μ m. **(also new Fig. 1d in the revised manuscript)**
- b. FRAP analysis showing fluorescence signal intensity recovery after photobleaching of LIN28A with and without the 1%HEX treatment; n=3 biologically independent experiments; Data are presented as mean values \pm SEM. SEM: standard error of the mean. **(also new Fig. 1e in the revised manuscript)**
- c. Representative confocal microscopy images of fluorescence recovery after photobleaching (FRAP) of the knock-in eGFP-LIN28A with and without the 1%HEX treatment in living WT cells. The targeted bleached region is highlighted with a white box. Scale bar, 5 μ m. **(also new Extended Data Fig.5a in the revised manuscript)**
- d. Statistical analysis of LIN28A loci area, LIN28A intensity and the irregularity of LIN28A in the nucleus with and without the 1%HEX treatment; n=20; *p < 0.05, **p < 0.01, ***p < 0.001, One-way ANOVA. **(also new Fig.1f,g,h in the revised manuscript)**

c. It is unclear how the authors performed FRAP experiments. Did the authors bleach half of the nucleolus, or the whole nucleolus? It is critical to explain, and to show the bleach images since to probe interior fluidity of the nucleolus, the authors should bleach half of it and analyze its recovery. If they bleached the whole, since there is not much LIN28A in the nucleoplasm, it is understandable that the recovery will be slow and not complete, because there is simply not much LIN28A surrounding the nucleolus to diffuse from. If this methodology is not established, any difference they see comparing WT and mutant LIN28A does not make much sense. It is also unclear FRAP in the cytoplasm is done. Are there cytoplasmic condensates of LIN28A too?

RESPONSE: We thank the reviewer for bringing our attention to this issue. For the FRAP experiments, we bleached half of the nucleolus or a portion of the nucleolus and showed the bleach images as below(**Response Figure 9a-9i**). Therefore, we believe we have employed the right methodology to evaluate the fluidity of nucleolus, because with our bleaching method the LIN28A protein can diffuse from the area outside of the bleached area, as the reviewer suggested.

In our experiment, we can see that LIN28A was generally diffusely distributed in the cytoplasm. We can't completely exclude there are some granules formed in the cytoplasm, such as the stress granules in certain stress conditions. FRAP experiments showed that LIN28A in the cytoplasm exhibited higher recovery compared to that in the nucleolus. Also WT type LIN28A and those LIN28A mutations showed different fluidity in the nucleolus, but had similar fluidity in the cytoplasm(**Response Figure 9g,9h**). Based on these reasons, we found it is more interesting to study the phase separated LIN28A in the nucleolus, therefore we focused on the nucleolar LIN28A condensates in this study.

Response Figure 9

Response Figure 9

- FRAP analysis showing FBL-mCherry fluorescence recovery after photobleaching in the indicated cells ($n=3$) in biologically independent experiments. Data are presented as mean values \pm SEM. SEM: standard error of the mean. (also Fig. 3e in the revised manuscript)
- FRAP analysis showing FBL fluorescence recovery after photobleaching in WT, *Lin28a* KO, and *Lin28a* KO cells transduced with Mut-LIN28A, S120A LIN28A, S200A LIN28A, R192G LIN28A, and FUS protein's IDR fused LIN28A variants ($n=3$) in biologically independent experiments. Data are presented as mean values \pm SEM. SEM: standard error of the mean. (also Fig. 4i in the revised manuscript)
- FRAP analysis showing NCL fluorescence recovery after photobleaching in the indicated cells ($n=3$) in biologically independent experiments. Data are presented as mean values \pm SEM. SEM standard error of the mean. (also Extended Data Fig.4c in the revised manuscript)
- Representative images of FBL fluorescence recovery after photobleaching (FRAP) in living WT, *Lin28a* KO, and *Lin28a* KO ESCs transduced with full length WT LIN28A or individual domain deleted LIN28A variants. Scale bar, 5 μ m. (also new Extended Data Fig.6a in the revised manuscript)
- Representative images of FBL fluorescence recovery after photobleaching (FRAP) in WT, *Lin28a* KO, and *Lin28a* KO cells transduced with Mut-LIN28A, S120A LIN28A, S200A LIN28A, R192G LIN28A, and FUS protein's IDR fused LIN28A variants cells. Scale bar, 5 μ m. (also new Extended Data Fig.6b in the revised manuscript)
- Representative images of NCL fluorescence recovery after photobleaching (FRAP) in the indicated cells. Scale bar, 5 μ m. (also new Extended Data Fig.6c in the revised manuscript)
- FRAP analysis showing WT-LIN28A, S120A, S200A, and R192G variants recovery after photobleaching in the cytoplasm and nucleolus ($n=3$) in biologically independent experiments. Data are presented as mean values \pm SEM. SEM: standard error of the mean. (also Fig.4e in the revised manuscript)
- Representative images of fluorescence recovery after photobleaching (FRAP) of WT-LIN28A, S120A and S200A, and R192G variants in the cytoplasm. Scale bar, 5 μ m. (also new Extended Data Fig.5e in the revised manuscript)

- i. FRAP analysis showing LIN28A fluorescence recovery after photobleaching in the WT nucleolus in the naïve state and the primed state (n=3) in biologically independent experiments. Data are presented as mean values \pm SEM. SEM standard error of the mean. **(also Extended Data Fig.9c in the revised manuscript)**
- j. FRAP analysis showing FBL fluorescence recovery after photobleaching in the WT nucleolus in the naïve state and the primed state (n=3) in biologically independent experiments. Data are presented as mean values \pm SEM. SEM standard error of the mean. **(also Extended Data Fig.9d in the revised manuscript)**
- k. Representative images of LIN28A fluorescence recovery after photobleaching (FRAP) in the WT nucleolus in the naïve state and the primed state. Scale bar, 5 μ m. **(also new Extended Data Fig.5f in the revised manuscript)**
- l. Representative images of FBL fluorescence recovery after photobleaching (FRAP) in the naïve state and the primed state. Scale bar, 5 μ m. **(also new Extended Data Fig.6d in the revised manuscript)**

d. The authors found most of the domains of LIN28A are important for proper organization of nucleolus. However, they decided to say that only IDRs are important, and are the ones they follow up with using individual mutations. This is a bit puzzling and needs better explanation.

RESPONSE: We thank the reviewer for bringing our attention to this issue. LIN28A contains two well-known RNA-binding domains (RBDs), a cold-shock domains (CSD), and a cysteine cysteine histidine cysteine (CCHC) zinc-finger domains (ZFD). A flurry of studies showing that LIN28A performed has an important role in reprogramming and maintenance of pluripotency roles through let-7 dependent (binding to let-7) or independent (binding directly to mature mRNA) pathways based on its RNA-binding domains^{4,6,24}.

We made the truncated mutants of RBDs and IDRs (**Figure 3**), and found that both RBDs and IDRs of LIN28A were important for proper organization of nucleolus. The RNA-binding domains have been studied intensively in the context of LIN28 function. We were more curious about the function of IDRs, which was previously assumed to have no functional roles in LIN28. In a protein is frequently assumed to be diagnostic of its ability to phase separate. Therefore, in this article, we focus on IDRs that have not been rarely intensively studied before.

To clarify this rationale for choosing IDR as the focus of this study, we have added more discussion in the Discussion section of the revised manuscript.

e. It is unclear if LIN28A leads to nucleolus formation, or is simply incorporated into the nucleolus. The authors seem to suggest that LIN28A and rRNA phase separation is sufficient to form the nucleolus (from the title of the paper and the in vitro data). But most likely LIN28A just incorporates into the existing nucleolus. If that's the case, then Figure 2 will not make much sense since depleting rRNA will only affect LIN28A phase separation indirectly by disrupting the nucleolus. The authors need to be less ambiguous about this important aspect of the paper.

RESPONSE: We thank the reviewer for bringing our attention to this issue. In terms of our current results, we should tune down our conclusion.

LIN28A is indeed integrated into nucleolus. LIN28A wraps around GC(NPM) in the nucleolus)(Figure 1c,6c in the revised manuscript). The cavities in the LIN28A inclusions tended to encapsulate DFC(FBL)(Figure 1b,1c in the revised manuscript). Loss of LIN28A resulted in disrupted stratification of nucleoli, but the disordered nucleoli were still present(Figure 5e in the revised manuscript). These results demonstrated LIN28A itself is not sufficient to form nucleolus, but its loss impairs the integrity of nucleolus. We can consider LIN28A as a marker of nucleolar integrity, it acts as a solid shell or scaffold to help stabilizing the existing nucleolar layered structure.

In Figure 2, we just used LIN28A as a marker of nucleolar integrity, and it's highly possible that disrupting rRNA indirectly affected LIN28A through first disrupting the nucleolar integrity. We also tune down the title from 'Dynamic Nucleolar Phase Separation Mediated by Non-canonical Function of LIN28A Instructs Pluripotent Stem Cell Fate Decisions' to 'Dynamic Nucleolar Phase Separation **Influenced** by Non-canonical Function of LIN28A Instructs Pluripotent Stem Cell Fate Decisions'.

2. A lot of essential data is also descriptive/only showing representative images, lacking quantifications. This is one of the main concerns of the paper. For example, the authors didn't use quantitative methods to show their data, but instead used descriptive and ambiguous ways of describing the nucleolar morphology (for example, using terms such as "floc", "diffused nucleoli", "fragmented nucleoli", "wreath structure"). These terms for a non-expert reader are hard to understand, and can potentially introduce biases when they quantify their images. Figures such as 2g, 3c, 4e, and 5h etc will benefit greatly from quantifications.

RESPONSE: We thank the reviewer for this valuable and constructive comment. As suggested, we have used quantitative methods to show the related data in our revised manuscript for the old Fig. 2g, 3c, 4e, and 5h, also shown below(Response Figure 10a-10y).

Response Figure 10

Response Figure 10

- a. Confocal microscopy Airyscan images of the morphology and nucleolar localization of LIN28A and FBL in the living control, CX-5461-treated, Pol I degraded, and snoRNA knockout mESCs. Scale bar, 5 μ m. **(also Fig. 2g in the revised manuscript)**
- b. Statistical analysis of LIN28A protein loci numbers in the nucleus in the indicated cells; n=33; *p < 0.05, **p < 0.01, ***p < 0.001, One-way ANOVA. **(also Fig. 2c in the revised manuscript)**
- c. Statistical analysis of LIN28A nucleus/cytoplasm fluorescence intensity ratios in the indicated cells; n=24; *p < 0.05, **p < 0.01, ***p < 0.001, One-way ANOVA. **(also Fig. 2d in the revised manuscript)**
- d. Statistical analysis of LIN28A loci area in the nucleus in the indicated cells; n=21; *p < 0.05, **p < 0.01, ***p < 0.001, One-way ANOVA. **(also Fig. 2e in the revised manuscript)**
- e. Statistical analysis of the numbers the typical morphology of FBL in the indicated cells; n=20 nucleoli; *p < 0.05, **p < 0.01, ***p < 0.001, One-way ANOVA. **(also new Fig. 2h in the revised manuscript)**
- f. Representative confocal microscopy Airyscan images of the morphology and nucleolar localization of LIN28A and FBL in living WT, Lin28a KO, LIN28A full-length LIN28A overexpressing, and truncated LIN28A overexpressing cells. Scale bar, 5 μ m. **(also Fig. 3c in the revised manuscript)**
- g. Statistical analysis of the numbers the typical morphology of FBL in the WT, Lin28a KO, LIN28A full-length LIN28A overexpressing, and truncated LIN28A overexpressing cells; n=20 nucleoli; *p < 0.05, **p < 0.01, ***p < 0.001, One-way ANOVA. **(also new Fig. 3d in the revised manuscript)**
- h. Representative confocal microscopy Airyscan images of the morphology and nucleolar localization of LIN28A and FBL in living WT, *Lin28a* KO, and *Lin28a* KO cells transduced with Mut-LIN28A, S120A LIN28A, S200A LIN28A, R192G LIN28A, and FUS protein's IDR fused LIN28A variants. Scale bar, 5 μ m. **(also Fig. 4f in the revised manuscript)**
- i. Statistical analysis of the numbers the typical morphology of FBL in living WT, *Lin28a* KO, and *Lin28a* KO cells transduced with Mut-LIN28A, S120A LIN28A, S200A LIN28A, R192G LIN28A, and FUS protein's IDR fused LIN28A variants; n=20; *p < 0.05, **p < 0.01, ***p < 0.001, One-way ANOVA. **(also new Fig. 4h in the revised manuscript)**
- j. Representative STED immunofluorescence images of LIN28A and NPM in MEF or iPSC cells. **(also new Fig. 6c in the revised manuscript)**
- k. Representative STED immunofluorescence images of NPM and FBL in MEF or iPSC cells. **(also new Fig. 6d in the revised manuscript)**
- l. Statistical analysis of the GC irregularity in MEF and iPSCs using Boyce-Clark semidiameter index. The larger the number, the more irregular the FBL. n=20 nucleoli; *p < 0.05, **p < 0.01, ***p < 0.001, Student's t-test, two-tailed. **(also new Fig. 6e in the revised manuscript)**
- m. Graph showing the number of typical DFC(FBL) morphology in MEF or iPSCs. n=20 nucleoli. **(also new Fig. 6f in the revised manuscript)**
- n. A cartoon diagram showing morphological changes of NPM and FBL in MEF or iPSC cells. **(also new Fig. 6g in the revised manuscript)**
- o. Representative images of the morphology and nucleolar localization of LIN28A and NCL in living ESCs of control cell, CX-5461 treatment cells, Pol I degraded cells, and snoRNA knockout cells. Scale bar, 5 μ m. **(also Extended Data Fig. 2b in the revised manuscript)**

- p. Statistical analysis of the numbers the typical morphology of NCL in living ESCs of control cell, CX-5461 treatment cells, Pol I degraded cells, and snoRNA knockout cells; n=20 nucleoli; *p < 0.05, **p < 0.01, ***p < 0.001, One-way ANOVA. **(also new Extended Data Fig.2c in the revised manuscript)**
- q. Statistical analysis of the inner diameter of NCL ring in living ESCs of control cell and CX-5461 treatment cells, n=20 nucleoli; *p < 0.05, **p < 0.01, ***p < 0.001, unpaired student's t-test, two-tailed. **(also new Extended Data Fig.2d in the revised manuscript)**
- r. Representative confocal microscopy images of the morphology and nucleolar localization of LIN28A and NCL in living WT, *Lin28a* KO, and *Lin28a* KO ESCs transduced with full length WT LIN28A or individual domain deleted LIN28A variants. Scale bar, 5µm. **(also Extended Data Fig.4b in the revised manuscript)**
- s. Statistical analysis of the numbers the typical morphology of NCL in the WT, *Lin28a* KO, LIN28A full-length LIN28A overexpressing, and truncated LIN28A overexpressing cells; n=20 nucleoli; *p < 0.05, **p < 0.01, ***p < 0.001, One-way ANOVA. **(also new Extended Data Fig.4d in the revised manuscript)**
- t. Immunofluorescence imaging showing OP-puromycin-labeled MEF and iPSCs. Scale bar, 20µm. **(also new Fig.6h in the revised manuscript)**
- u. Statistical analysis of the OP-Puro intensity in MEF and iPSCs. n=20 cells; *p < 0.05, **p < 0.01, ***p < 0.001, Student's t-test, two-tailed. **(also new Fig.6h in the revised manuscript)**
- v. Fluorescence imaging showing OP-puromycin-labeled WT ESCs in naïve and primed state. Scale bar, 100µm. **(also new Extended Data Fig.9a in the revised manuscript)**
- w. Statistical analysis of the OP-Puro intensity in WT ESCs in naïve and primed state. n=20 cells; *p < 0.05, **p < 0.01, ***p < 0.001, unpaired two-tailed student's t-test. **(also new Extended Data Fig.9a in the revised manuscript)**
- x. Fluorescence imaging showing OP-puromycin-labeled *Lin28a* KO ESCs in naïve and primed state. Scale bar, 100µm. **(also new Extended Data Fig.9b in the revised manuscript)**
- y. Statistical analysis of the OP-Puro intensity in *Lin28a* KO ESCs in naïve and primed state. n=20 cells; *p < 0.05, **p < 0.01, ***p < 0.001, unpaired two-tailed student's t-test. **(also new Extended Data Fig.9b in the revised manuscript)**

3. The article seems to compile a number of experiments without clear explanation of their results. The article as a whole has a lot of results that are not discussed very well, or seems out of place. For example, the cold-responsiveness of LIN28A phase separation seems out of place. I am also not sure what it means to have a slower FRAP recovery after hex treatment. The authors need to connect and discuss their data instead of simply presenting them.

RESPONSE: We thank the reviewer for bringing our attention on this issue. We agree with this suggestion that the cold-responsiveness of LIN28A phase separation seems out of place. We have removed it in the revised manuscript. .

Mammalian nucleolus was a multiphase liquid condensate. The aliphatic alcohol 1,6-hexanediol (HEX) interferes with weak hydrophobic interactions and is often used to dissolve protein condensates in cells²³. LIN28A is present both in the nucleolus and cytoplasm. We can use HEX to study in which compartment can LIN28A form condensate. Confocal imagings and the FRAP experiment indicated that HEX treatment did not affect cytoplasmic LIN28A distribution, suggesting that cytoplasmic LIN28A was more diffused and did not have typical phase-separated condensate behavior. However, the nucleolar LIN28A was disrupted and the fluidity slowed considerably after HEX treatment, suggesting that nucleolar LIN28A assumed the phase-separated condensate features.

We have added this rationale in the revised manuscript:

Line93-107: “Mammalian nucleolus was a multiphase liquid condensate. The aliphatic alcohol 1,6-hexanediol (HEX) interferes with weak hydrophobic interactions and is often used to dissolve protein condensates in cells. LIN28A is present both in the nucleolus and cytoplasm. Thus we used HEX to study in which compartment can LIN28A form condensate. Endogenous eGFP-LIN28A knock-in ESCs treated with 1% HEX for 10 minutes showed diffusion of the condensates in the nucleolus (Fig. 1d). To test the fluidity of LIN28A protein condensates, we performed FRAP experiments and found that LIN28A in the nucleolus exhibited slower recovery compared to that in the cytoplasm, and the nucleolar LIN28A condensate was more sensitive to the HEX treatment (Fig. 1e and Extended Data Fig. 5a). The statistical analysis also quantitatively showed a reduction in LIN28A condensates intensity in the nucleolus after HEX treatment (Fig. 1f), and an increase in the dispersed area and irregularity of LIN28A in the nucleolus after HEX treatment (Fig. 1g,h). HEX treatment did not affect cytoplasmic LIN28A distribution, suggesting that cytoplasmic LIN28A was more diffused and did not have typical phase-separated condensate behavior. In contrast, the nucleolar LIN28A was disrupted and the fluidity slowed considerably after HEX treatment, suggesting that nucleolar LIN28A assumed the phase-separated condensate features.”

Other points:

1. Lines 104-105: data in Fig. 2f doesn't support the claim that LIN28A expression level doesn't change: there is a decrease in LIN28A at 42°C. GAPDH blot also seems to be cropped/manipulated.

RESPONSE: We thank the reviewer for focusing our attention on this issue. We decided to remove the temperature-related text after careful consideration as the cold-responsiveness of LIN28A phase separation seems out of place.

2. Fig. 1i: separation of LIN28A from FBL doesn't automatically mean it colocalizes with other markers of GC and DFC. To demonstrate this, the authors need to show colocalization of LIN28A with GC and DFC.

RESPONSE: We thank the reviewer for this good critic. Following the reviewer's suggestion, we have investigated the colocalization of LIN28A with the widely used DFC marker protein FBL, and GC marker protein NPM by STED. (also new **Fig.1c** in the revised manuscript)

3. Line 128: in vivo should be changed to in cell.

RESPONSE: We thank the reviewer for focusing our attention on this issue. According to the reviewer's suggestion, we corrected this description in our revised manuscript.

Line 122 : “rRNA is essential to maintain the localization and fluidity of LIN28A in the nucleolus”

4. Line 199: the authors didn't include the in vitro data showing poor solubility and prone to aggregation.

RESPONSE: We thank the reviewer for focusing our attention on this issue. We removed this description in our revised manuscript.

References

- 1 Ambros, V. & Horvitz, H. R. Heterochronic mutants of the nematode *Caenorhabditis elegans*. *Science* **226**, 409-416 (1984).
- 2 Rybak, A. *et al.* A feedback loop comprising *lin-28* and *let-7* controls pre-*let-7* maturation during neural stem-cell commitment. *Nat Cell Biol* **10**, 987-993, doi:10.1038/ncb1759 (2008).
- 3 Heo, I. *et al.* *Lin28* mediates the terminal uridylation of *let-7* precursor MicroRNA. *Mol Cell* **32**, 276-284, doi:10.1016/j.molcel.2008.09.014 (2008).
- 4 Viswanathan, S. R. & Daley, G. Q. *Lin28*: A microRNA regulator with a macro role. *Cell* **140**, 445-449, doi:10.1016/j.cell.2010.02.007 (2010).
- 5 Zhang, J. *et al.* *LIN28* Regulates Stem Cell Metabolism and Conversion to Primed Pluripotency. *Cell Stem Cell* **19**, 66-80, doi:10.1016/j.stem.2016.05.009 (2016).
- 6 Nam, Y., Chen, C., Gregory, R. I., Chou, J. J. & Sliz, P. Molecular basis for interaction of *let-7* microRNAs with *Lin28*. *Cell* **147**, 1080-1091, doi:10.1016/j.cell.2011.10.020 (2011).
- 7 Cho, J. *et al.* *LIN28A* is a suppressor of ER-associated translation in embryonic stem cells. *Cell* **151**, 765-777 (2012).
- 8 Hafner, M. *et al.* Identification of mRNAs bound and regulated by human *LIN28* proteins and molecular requirements for RNA recognition. *Rna* **19**, 613-626 (2013).
- 9 Lu, C. & Liu, Y. Effects of China's urban form on urban air quality. *Urban Studies* **53**, 2607-2623 (2016).
- 10 Mazid, M. A. *et al.* Rolling back human pluripotent stem cells to an eight-cell embryo-like stage. *Nature* **605**, 315-324, doi:10.1038/s41586-022-04625-0 (2022).
- 11 Bi, Y. *et al.* Cell fate roadmap of human primed-to-naïve transition reveals preimplantation cell lineage signatures. *Nature Communications* **13**, doi:10.1038/s41467-022-30924-1 (2022).
- 12 Ai, Z. *et al.* Krüppel-like factor 5 rewires NANOG regulatory network to activate human naïve pluripotency specific LTR7Ys and promote naïve pluripotency. *Cell Reports* **40**, doi:10.1016/j.celrep.2022.111240 (2022).
- 13 Messmer, T. *et al.* Transcriptional Heterogeneity in Naïve and Primed Human Pluripotent Stem Cells at Single-Cell Resolution. *Cell Reports* **26**, 815-824.e814, doi:10.1016/j.celrep.2018.12.099 (2019).
- 14 Monahan, Z. *et al.* Phosphorylation of the FUS low-complexity domain disrupts phase separation, aggregation, and toxicity. *EMBO J* **36**, 2951-2967, doi:10.15252/embj.201696394 (2017).
- 15 Shin, Y. *et al.* Spatiotemporal Control of Intracellular Phase Transitions Using Light-Activated optoDroplets. *Cell* **168**, 159-171 e114, doi:10.1016/j.cell.2016.11.054 (2017).
- 16 Wang, J. *et al.* Phase separation of OCT4 controls TAD reorganization to promote cell fate transitions. *Cell Stem Cell* **28**, 1868-1883 e1811, doi:10.1016/j.stem.2021.04.023 (2021).
- 17 Tsanov, K. M. *et al.* *LIN28* phosphorylation by MAPK/ERK couples signalling to the post-transcriptional control of pluripotency. *Nat Cell Biol* **19**, 60-67, doi:10.1038/ncb3453 (2017).
- 18 Smith, A. G. *et al.* Inhibition of pluripotential embryonic stem cell differentiation by purified polypeptides. *nature* **336**, 688-690 (1988).
- 19 Williams, R. L. *et al.* Myeloid leukaemia inhibitory factor maintains the developmental potential of embryonic stem cells. *Nature* **336**, 684-687 (1988).
- 20 Graf, U., Casanova, E. A. & Cinelli, P. The Role of the Leukemia Inhibitory Factor (LIF) — Pathway in Derivation and Maintenance of Murine Pluripotent Stem Cells. *Genes* **2**, 280-297, doi:10.3390/genes2010280 (2011).
- 21 Cherepkova, M. Y., Sineva, G. S. & Pospelov, V. A. Leukemia inhibitory factor (LIF) withdrawal activates mTOR signaling pathway in mouse embryonic stem cells through the MEK/ERK/TSC2 pathway. *Cell death & disease* **7**, e2050-e2050 (2016).
- 22 Mummery, C., Feyen, A., Freund, E. & Shen, S. Characteristics of embryonic stem cell differentiation: a comparison with two embryonal carcinoma cell lines. *Cell Differentiation and Development* **30**, 195-206 (1990).
- 23 Duster, R., Kaltheuner, I. H., Schmitz, M. & Geyer, M. 1,6-Hexanediol, commonly used to dissolve liquid-liquid phase separated condensates, directly impairs kinase and phosphatase activities. *J Biol Chem* **296**, 100260, doi:10.1016/j.jbc.2021.100260 (2021).
- 24 Wu, K., Ahmad, T. & Eri, R. *LIN28A*: A multifunctional versatile molecule with future therapeutic potential. *World Journal of Biological Chemistry* **13**, 35-46, doi:10.4331/wjbc.v13.i2.35 (2022).

REVIEWER COMMENTS

Reviewer #1 (Remarks to the Author):

The authors did experiments to answer some questions. Questions are remaining to be answered. And the data should be clear explained and connected to be form a story.

1, It is important to clarify the difference of the phase separation and other functions of LIN28A. In the revision, the authors only test the microRNA let-7g and one mRNA in the IDR mutation. This can not rule out other RNAs that can be affected by these IDR mutations. Also the authors conclude that phosphor-null LIN28A lose phase separation property and capacity of cell fate. Here the difference of the phase separation and other functions of LIN28A also need to be clarified. The whole genome-wide Seq is suggested to investigate the RNA binding and regulation and sequential functions.

2, It is still unclear whether and how cytoplasmic condensates of LIN28A, and the interaction and molecular linking mechanism between nucleolar and cytoplasmic condensates of LIN28A. Suggest more data to elucidate the cytoplasmic and nucleolar condensates of LIN28A, which one loses the functions of RNA binding, and function of mouse/human iPSC and mouse naïve-to-primed function. The colocalization of LIN28A is puzzling. The results in the revision is controversial from the original data. The solution of image is not the level of supersolution of STED, but as the normal fluorescent microscopy. Fig 1C ,7, LIN28A is oversaturated, could not draw proper conclusion. Suggest for cytosol and nucleolar one, the real supersolution image, 3D image and/or immuno-EM to show the clear localization of both overexpressed and endogenous LIN28A.

3, The function of the phase separation of LIN28A in human pluripotent stem cell differentiation is still lacking. And instead of carefully investigating the temperature regulating LIN28A, the authors directly delete this part. Suggest further investigation and data to clearly show the hot and cold condition on LIN28A expression and phase separation.

4,It is interesting to find that both RBDs and IDRs of LIN28A were important for proper organization of nucleolus. However, the authors only focused on IDRs, which is well known important for phase separation, and ignore RBDs. Suggest mutations of RBDs in overexpression and knock-in system to investigate its role. And the mechanism of LIN28A leading to nucleolus formation or simply being incorporated into the nucleolus is still unclear. The relationship between LIN28A, rRNA and constructive nucleolus protein machine needs to be investigated. Besides in vivo system, the in vitro nucleosome assembly structure by EM and physical quantity is suggested to answer this.

Minor:

The added data such as Fig 2h,3d,4ghi, 6f,S2c S4d, lacking statistics. The western and also other data also lacking statistics. Three or more independent biological repeats should be required.

Reviewer #2 (Remarks to the Author):

In this revised manuscript, the authors included a lot of new data, including the LIN28 CRISPR KI cell line and image data quantifications. I have a few more comments before the manuscript can be accepted:

1) There is still confusion here: "However, the nucleolar LIN28A was disrupted and the fluidity slowed considerably after HEX treatment, suggesting that nucleolar LIN28A assumed the phase-separated condensate features." Do the authors mean that LIN28A only assumed condensate features before or after Hex treatment? The authors need to be a bit clearer in their writing. I also don't get the logic that slower FRAP recovery after Hex shows it is phase separated. There is no basis for claiming this. Usually Hex is just used to dissolve the condensates. I would suggest to just delete the Hex data if it is not relevant elsewhere.

2) There are more "in vivo" in the text that needs to be changed to "in cell". "In vivo" is usually used to describe in an animal.

We thank the reviewers for the questions and have address them in the point-to-point letter as below.

Reviewer #1 (Remarks to the Author):

The authors did experiments to answer some questions. Questions are remaining to be answered. And the data should be clear explanted and connected to be form a story.

1, It is important to clarify the difference of the phase separation and other functions of LIN28A. In the revision, the authors only test the microRNA let-7g and one mRNA in the IDR mutation. This can not rule out other RNAs that can be affected by these IDR mutations. Also the authors conclude that phosphor-null LIN28A lose phase separation property and capacity of cell fate. Here the difference of the phase separation and other functions of LIN28A also need to be clarified. The whole genome-wide Seq is suggested to investigate the RNA binding and regulation and sequential functions.

RESPONSE: We thank the reviewer for focusing our attention on this issue. In our manuscript, LIN28A IDR single amino acid mutants and FUS IDR fusion rescue experiments have shown phosphor-null LIN28A lose phase separation property and capacity of cell fate (**Fig4-7**). About “This can not rule out other RNAs that can be affected by these IDR mutations, and the difference of the phase separation and other functions of LIN28A also need to be clarified”, our response is as follows:

In mouse and human, the let-7 family miRNAs comprise 12 members, the mature miRNA sequence of which is highly conserved between the different genes. Let-7 is certainly a key target of LIN28A. Mouse Lin28A has two folded RNA binding domains, the CSD and the CCHC₂. The Lin28A CSD and the CCHC “zinc-finger domains” (ZFD) make extensive contacts with the pre-let7 elements in two distinct regions. Lin28 ZFD specifically recognizes a conserved GGAG motif within pre-let-7. Neither the terminal nor linker regions outside of the folded domains are essential for blocking let-7¹. Our EMSA results (Shown in **Extended Data Fig.6 in the revised manuscript**) are consistent with previous studies. A number of genome-wide Lin28 RNA crosslinking and immunoprecipitation coupled to high-throughput sequencing (HITS-Clip and PAR-CLIP) studies were conducted in human and mouse ESCs²⁻⁴. mRNA is a major class of LIN28A targets. LIN28A acted as post-transcriptional regulator of mRNA translation, and the activity strongly depended on it’s two RNA-binding domains⁵. In conclusion, our LIN28A IDR single amino acid mutants just changed nucleolar phase separation, but did not affected its RNA-binding capacity in the cytoplasm.

2, It is still unclear whether and how cytoplasmic condensates of LIN28A, and the interaction and molecular linking mechanism between nucleolar and cytoplasmic condensates of LIN28A. Suggest more data to elucidate the cytoplasmic and nucleolar condensates of LIN28A, which one loses the functions of RNA binding, and function of mouse/human iPSC and mouse naïve-to-primed function. The colocalization of LIN28A is puzzling. The results in the revision is controversial from the original data. The solution of image is not the level of supersolution of STED, but as the normal fluorescent microscopy. Fig 1C ,7, LIN28A is oversaturated, could not draw proper conclusion. Suggest for cytosol and nucleolar one, the real supersolution image, 3D image and/or immuno-EM to show the clear localization of both overexpressed and endogenous LIN28A.

RESPONSE: We thank the reviewer for focusing our attention on this issue. We have broken down the questions to address each of them, and also address how we come to our conclusion in the following point-to-point manner:

a In the context of our result, we can see LIN28A IDR single amino acid mutants did not change

cytoplasmic location and fluidity. These mutants just changed their nucleolar localization and fluidity (Shown in **Fig.4d-h in the revised manuscript**). These IDR mutations did not affect LIN28A's canonical functions such as RNA binding, at least for the microRNA let-7g and the OxPhos gene *Ndufb10* mRNA as we tested. **In other words, alterations in a region that promotes nucleolar LIN28A phase-separated condensates did not alter the functions of LIN28A in the cytoplasm in terms of cytoplasmic RNA binding.**

- b For the question about whether it is cytoplasmic or nucleolar condensates of LIN28A that mediates the function of RNA binding or naïve-to-primed transition, we have shown that the nucleolar LIN28A mediates the naïve-to-primed transition and does not involve cytoplasmic let-7 RNA binding to mediate the naïve-to-primed transition. Specifically, we have shown the single amino acids mutation in the LIN28A' IDR region affect its morphology and phase separation capacity in the **nucleolus**, and did not affect its distribution and fluidity in the **cytoplasm**. Besides, the fused FUS IDRs rescued the morphology and phase separation capability of the LIN28A mutants in the **nucleolus** (Shown in **Fig.4d, f, h in the revised manuscript**). Meanwhile, the three IDR fusions completely rescued the impaired morphology and fluidity of FBL caused by the LIN28A mutants in the **nucleolus** (Shown in **Fig.4k, l in the revised manuscript**). Also, for the mouse embryonic fibroblast (MEF) cells and neonatal human dermal fibroblast (NHDF) cells reprogrammed by OCT4, SOX2, NANOG and LIN28A, LIN28A IDR single amino acid mutations led to reduced reprogramming efficiency, which can be rescued by fusion of FUS IDRs (Shown in **Fig. 6a, b and Fig. 7b, c in the revised manuscript**), demonstrating phase separation of **nucleolar** LIN28A facilitated reprogramming.
- c Similarly, for the phenomenon that three LIN28A IDR single amino acid mutations caused naïve-to-primed transition defects, **nucleolar LIN28A phase separation, BUT NOT cytoplasmic LIN28A** played a crucial role in this process, as the IDR mutations only affect nucleolar LIN28A, but not cytoplasmic LIN28A, as described above.
- d With respect to the issue of “Fig 1C ,7, LIN28A is oversaturated, could not draw proper conclusion”. We have replaced the images without oversaturation accordingly.
- e With respect to the issue of “The colocalization of LIN28A is puzzling. The results in the revision is controversial from the original data. The solution of image is not the level of supersolution of STED, but as the normal fluorescent microscopy. Suggest for cytosol and nucleolar one, the real supersolution image, 3D image and/or immuno-EM to show the clear localization of both overexpressed and endogenous LIN28A”. Immunostaining by Stimulated Emission Depletion Microscopy(STED) showed co-localization of LIN28A with DFC(FBL) and GC(NPM) in the nucleolus in E14 mESCs cultured in LIF/serum. LIN28A covered a larger region around FBL with empty holes in the middle, consistent with previous study that LIN28A was distributed mainly in the GC and DFC regions⁶ (Shown in **Fig. 1c in the revised manuscript**). Our STED images (STED Resol. applied for FBL and NPM) were acquired using Abberior Instruments with z-stack module. The x, y, and z axis resolution was 30nm. STED Resol. was 5% (Shown in **Fig. a-h**).

We also compared the resolution of our images of supersolution STED or SIM with the previously published paper^{7,8} shown as below. Our images above also illustrated the fine structure of nucleolus as the previous publications as below.

[REDACTED]

3, The function of the phase separation of LIN28A in human pluripotent stem cell differentiation is still lacking. And instead of carefully investigating the temperature regulating LIN28A, the authors directly delete this part. Suggest further investigation and data to clearly show the hot and cold condition on LIN28A expression and phase separation.

RESPONSE: We thank the reviewer for focusing our attention on this issue. We have shown the hot and cold condition on LIN28A expression and phase separation.

(Shown in **Extended Data Fig.1e-j** in the revised manuscript)

4, It is interesting to find that both RBDs and IDRs of LIN28A were important for proper organization of nucleolus. However, the authors only focused on IDRs, which is well known important for phase separation, and ignore RBDs. Suggest mutations of RBDs in overexpression and knock-in system to investigate its role. And the mechanism of LIN28A leading to nucleolus formation or simply being incorporated into the nucleolus is still unclear. The relationship between LIN28A, rRNA and constructive nucleolus protein machine needs to be investigated. Besides in vivo system, the in vitro nucleosome

assembly structure by EM and physical quantity is suggested to answer this.

RESPONSE: We thank the reviewer for focusing our attention on this issue. With respect to the issue of “It is interesting to find that both RBDs and IDRs of LIN28A were important for proper organization of nucleolus. However, the authors only focused on IDRs, which is well known important for phase separation, and ignore RBDs. Suggest mutations of RBDs in overexpression and knock-in system to investigate its role”. A flurry of studies showed that LIN28A had an important role in reprogramming and maintenance of pluripotency roles through let-7 dependent (binding to let-7) or independent (binding directly to mature mRNA) pathways based on its RNA-binding domains^{1,9,10}. We made the truncated mutants of RBDs and IDRs (**Figure 3**). While we found that both RBDs and IDRs of LIN28A were important for proper organization of nucleolus, the RNA-binding domains have been studied intensively in the context of LIN28 function. We were more curious about the function of IDRs, which was previously assumed to have no functional roles in LIN28A. Therefore, in this article, we focus on IDRs that have been rarely studied before. To clarify this rationale for choosing IDR as the focus of this study, we have added more discussion in the Discussion section of the revised manuscript.

With respect to the issue of “And the mechanism of LIN28A leading to nucleolus formation or simply being incorporated into the nucleolus is still unclear. The relationship between LIN28A, rRNA and constructive nucleolus protein machine needs to be investigated. Besides in vivo system, the in vitro nucleosome assembly structure by EM and physical quantity is suggested to answer this”. LIN28A is indeed integrated into nucleolus. LIN28A wraps around GC(NPM) in the nucleolus (**Figure 1c,6c in the revised manuscript**). The cavities in the LIN28A inclusions tended to encapsulate DFC(FBL) (**Figure 1b,1c in the revised manuscript**). Loss of LIN28A resulted in disrupted stratification of nucleoli, but the disordered nucleoli were still presented (**Figure 5e in the revised manuscript**). These results demonstrated LIN28A itself is not sufficient to form nucleolus, but its loss impairs the integrity of nucleolus. We can consider LIN28A as a marker of nucleolar integrity, it acts as a solid shell or scaffold to help stabilizing the existing nucleolar layered structure. In Figure 2, we can see the nucleolar LIN28A condensate was sensitive to the rRNA inhibition, and it could be used as a marker of nucleolar integrity. It’s highly possible that disrupting rRNA indirectly affected LIN28A through first disrupting the nucleolar integrity.

Minor:

The added data such as Fig 2h,3d,4ghi, 6f,S2c S4d, lacking statistics. The western and also other data also lacking statistics. Three or more independent biological repeats should be required.

RESPONSE: We thank the reviewer for focusing our attention on this issue. We have performed statistical analysis on the above mentioned data Fig 2h,3d,4ghi, 6f, S2c S4d, and p values were labeled in the figure legend. We have performed quantitative or statistical analysis on the western (Shown in **Extended Data Fig.3d-f in the revised manuscript**) and EMSA data (Shown in **Extended Data Fig.6d in the revised manuscript**). The data comes from three independent biological repeats.

Reviewer #2 (Remarks to the Author):

In this revised manuscript, the authors included a lot of new data, including the LIN28 CRISPR KI cell line and image data quantifications. I have a few more comments before the manuscript can be accepted: 1) There is still confusion here: "However, the nucleolar LIN28A was disrupted and the fluidity slowed considerably after HEX treatment, suggesting that nucleolar LIN28A assumed the phase-separated condensate features." Do the authors mean that LIN28A only assumed condensate features before or after Hex treatment? The authors need to be a bit clearer in their writing. I also don't get the logic that slower FRAP recovery after Hex shows it is phase separated. There is no basis for claiming this. Usually Hex is just used to dissolve the condensates. I would suggest to just delete the Hex data if it is not relevant elsewhere.

RESPONSE: We thank the reviewer for focusing our attention on this issue. The aliphatic alcohol 1,6-hexanediol (HEX) interferes with weak hydrophobic interactions and is often used to dissolve liquid–liquid phase separated condensates¹¹. LIN28A is present both in the nucleolus and cytoplasm. We used HEX to prove in which compartment can LIN28A form liquid–liquid phase separated condensates. We utilized FRAP to confirm that LIN28A molecular fluidity characteristics in cells. Fluidity is not equal to phase separation capacity. According to the reviewer's suggestion, we delete the related Hex data.

2) There are more "in vivo" in the text that needs to be changed to "in cell". "In vivo" is usually used to describe in an animal.

RESPONSE: We thank the reviewer for focusing our attention on this issue. According to the reviewer's suggestion, we corrected this description in our revised manuscript.

- 1 Nam, Y., Chen, C., Gregory, R. I., Chou, J. J. & Sliz, P. Molecular basis for interaction of let-7 microRNAs with Lin28. *Cell* **147**, 1080-1091, doi:10.1016/j.cell.2011.10.020 (2011).
- 2 Hafner, M. *et al.* Identification of mRNAs bound and regulated by human LIN28 proteins and molecular requirements for RNA recognition. *Rna* **19**, 613-626, doi:10.1261/rna.036491.112 (2013).
- 3 Cho, J. *et al.* LIN28A Is a Suppressor of ER-Associated Translation in Embryonic Stem Cells. *Cell* **151**, 765-777, doi:10.1016/j.cell.2012.10.019 (2012).
- 4 Graf, R. *et al.* Identification of LIN28B-bound mRNAs reveals features of target recognition and regulation. *RNA biology* **10**, 1146-1159, doi:10.4161/rna.25194 (2013).
- 5 Mayr, F. & Heinemann, U. Mechanisms of Lin28-mediated miRNA and mRNA regulation--a structural and functional perspective. *Int J Mol Sci* **14**, 16532-16553, doi:10.3390/ijms140816532 (2013).
- 6 Sun, Z. *et al.* LIN28 coordinately promotes nucleolar/ribosomal functions and represses the 2C-like transcriptional program in pluripotent stem cells. *Protein Cell* **13**, 490-512, doi:10.1007/s13238-021-00864-5 (2022).
- 7 Yao, R. W. *et al.* Nascent Pre-rRNA Sorting via Phase Separation Drives the Assembly of Dense Fibrillar Components in the Human Nucleolus. *Mol Cell* **76**, 767-783 e711, doi:10.1016/j.molcel.2019.08.014 (2019).
- 8 Wu, M. *et al.* lncRNA SLERT controls phase separation of FC/DFCs to facilitate Pol I transcription. *Science* **373**, 547-555, doi:doi:10.1126/science.abf6582 (2021).
- 9 Viswanathan, S. R. & Daley, G. Q. Lin28: A microRNA regulator with a macro role. *Cell* **140**, 445-449, doi:10.1016/j.cell.2010.02.007 (2010).

- 10 Wu, K., Ahmad, T. & Eri, R. LIN28A: A multifunctional versatile molecule with future therapeutic potential. *World Journal of Biological Chemistry* **13**, 35-46, doi:10.4331/wjbc.v13.i2.35 (2022).
- 11 Duster, R., Kaltheuner, I. H., Schmitz, M. & Geyer, M. 1,6-Hexanediol, commonly used to dissolve liquid-liquid phase separated condensates, directly impairs kinase and phosphatase activities. *J Biol Chem* **296**, 100260, doi:10.1016/j.jbc.2021.100260 (2021).

REVIEWERS' COMMENTS

Reviewer #1 (Remarks to the Author):

The authors have used the mouse system to explore the function of the phase separation of LIN28A, whose IDR domain plays important roles. The human system or other domain work is not required. There are some questions remaining to be answered.

1, The authors conclude that phosphor-null LIN28A loses phase separation property and capacity of cell fate. It has been reported that this modification of LIN28A plays roles in mRNA targeting. Here the difference of the phase separation and mRNA targeting of LIN28A phosphorylation need to be clarified by data.

2, The authors showed LIN28A IDR single amino acid mutants did not change cytoplasmic location and fluidity by imaging to rule out the cytoplasmic LIN28A. Besides imaging, suggest to do isolation of nucleoids and cytoplasm to do WB to see the amount/length, clarify the possible LIN28A translocation out or in the nucleoids or posttranslation to result the different localization of LIN28A, based on which to further see the interaction and molecular linking mechanism between nucleolar and cytoplasmic condensates of LIN28A.

3, Current STED image does not show the detailed ultra-structure of LIN28A in nucleosomes. Besides the STED, suggest to observe the ultra-structure of LIN28A in nucleosomes by using such as PMID25768910/37816746 STORM/cryo-ET.

4, The temperature regulating LIN28A is interesting. Besides imaging cells, suggest in vitro phase separation data to add. And what is the mechanism of the same and difference between the cytoplasmic and nuclear LIN28A response to temperature, and the linkage between them?

5, The mechanism of LIN28A leading to nucleolus formation or simply being incorporated into the nucleolus is still unclear. The relationship between LIN28A, rRNA and constructive nucleolus protein machine needs to be investigated. Besides in vivo system, the in vitro nucleosome assembly structure by EM and physical quantity is suggested to answer this.

Reviewer #2 (Remarks to the Author):

I have no further comments for the paper.

We thank the reviewers for the questions and have address them in the point-to-point letter as below.

Reviewer #1 (Remarks to the Author):

The authors have used the mouse system to explore the function of the phase separation of LIN28A, whose IDR domain plays important roles. The human system or other domain work is not required. There are some questions remaining to be answered.

1, The authors conclude that phosphor-null LIN28A loses phase separation property and capacity of cell fate. It has been reported that this modification of LIN28A plays roles in mRNA targeting. Here the difference of the phase separation and mRNA targeting of LIN28A phosphorylation need to be clarified by data.

RESPONSE: We thank the reviewer for focusing our attention on this issue. LIN28A acts as post-transcriptional regulator of mRNA translation, and the activity strongly depends on it's two RNA-binding domains.

Our results have demonstrated that the S120A (Located in the linker disordered region), S200A (Located in C-terminal disordered regions) phosphor-null LIN28A were essential for its phase separation function in the nucleolus, as well as for maintaining normal nucleolar integrity. S120A and S200A IDR mutations did not affect LIN28A's canonical functions such as RNA binding and regulation, at least for the microRNA let-7g and the *Ndufb10* mRNA we tested (**Supplementary Fig.6c,d and Response Fig.1d**).

Yet it is worth noting that our study can not completely rule out there are other RNAs that can be affected by these phosphor-null LIN28A mutations. This is a good question to explore in future studies. We have added this sentence at the line 271-272 of page 10.

2, The authors showed LIN28A IDR single amino acid mutants did not change cytoplasmic location and fluidity by imaging to rule out the cytoplasmic LIN28A. Besides imaging, suggest to do isolation of nucleoids and cytoplasm to do WB to see the amount/length, clarify the possible LIN28A translocation out or in the nucleoids or posttranslation to result the different localization of LIN28A, based on which to further see the interaction and molecular linking mechanism between nucleolar and cytoplasmic condensates of LIN28A.

RESPONSE: We thank the reviewer for focusing our attention on this issue. Here imaging is the appropriate experiment because DAPI was used to allow us to distinguish nucleolus and cytosol. If using nucleolus and cytosol isolation ,we would not be able to see the morphology of the condensed proteins because they will be completely disrupted.. Using imaging approach, we have successfully investigated the colocalization of LIN28A with the widely used DFC marker protein FBL, and GC marker protein NPM by STED (**Fig.1c in the revised manuscript**).

We have also showed LIN28A IDR single amino acid mutants did not change cytoplasmic location and fluidity by imaging to rule out the cytoplasmic LIN28A. We did not intend to study the molecular linking mechanism between nucleolar and cytoplasmic LIN28A, nor the translocation of LIN28A, which is an interesting question but beyond the scope of the current study.

3, Current STED image does not show the detailed ultra-structure of LIN28A in nucleosomes. Besides the STED, suggest to observe the ultra-structure of LIN28A in nucleosomes by using such as

PMID25768910/37816746 STORM/cryo-ET.

RESPONSE: We thank the reviewer for focusing our attention on this issue. We have studied the method mentioned in PMID25768910/37816746. Nucleosomes help the formation of chromosomes by compacting DNA into fibers¹. The chromatin fibres are not structured as uniform 30nm one-start or two-start filaments but are composed of relaxed, variable zigzag organizations of nucleosomes connected by straight linker DNA². In this paper (PMID25768910), STORM showed increased levels of H1 in larger and denser clutches containing more nucleosomes, which formed the “closed” heterochromatin. On the other hand, “open” chromatin was formed by smaller and less dense clutches which associated with RNA Polymerase II.

LIN28A overlaps with the granular component (GC, labeled by NPM1) where ribosome assembly takes place, and with the inner dense fibrillar component (DFC, marked by Fibrillan or FBL) where rRNA is modified, and has the least overlap with the core of fibrillar center (FC, marked by RNA Pol I)³ where rDNA is present. STED observed co-localization of LIN28A with DFC(FBL) and GC(NPM) in the nucleolus in mESCs. LIN28A overlaps with the granular component and covered a larger region around DFC with empty holes in the middle. We think that LIN28A mainly plays a role in the establishment of DFC and GC in the nucleus, but not involved in the nucleosome. Thus this question is out of scope of our present study, we can explore in future studies.

4, The temperature regulating LIN28A is interesting. Besides imaging cells, suggest *in vitro* phase separation data to add. And what is the mechanism of the same and difference between the cytoplasmic and nuclear LIN28A response to temperature, and the linkage between them?

RESPONSE: We thank the reviewer for focusing our attention on this issue. Previous studies on phase separation have demonstrated that temperature influences droplet formation *in vitro*⁴. LIN28A, as a phase separation protein, formed droplets faster when incubated at 37 ° C than at room temperature in the course of our experiment (It's not shown here). With respect to the issue of “The mechanism of the same and difference between the cytoplasmic and nuclear LIN28A response to temperature, and the linkage between them”, as in question #2, we did not intend to study the molecular linking mechanism between nucleolar and cytoplasmic LIN28A, nor the translocation of LIN28A, which is an interesting question but beyond the scope of the current study. We can explore it in future studies.

5, The mechanism of LIN28A leading to nucleolus formation or simply being incorporated into the nucleolus is still unclear. The relationship between LIN28A, rRNA and constructive nucleolus protein machine needs to be investigated. Besides *in vivo* system, the *in vitro* nucleosome assembly structure by EM and physical quantity is suggested to answer this.

RESPONSE: We thank the reviewer for focusing our attention on this issue. With respect to the issue of “The mechanism of LIN28A leading to nucleolus formation or simply being incorporated into the nucleolus is still unclear. The relationship between LIN28A, rRNA and constructive nucleolus protein machine needs to be investigated”. LIN28A is indeed integrated into nucleolus. LIN28A wraps around GC(NPM) in the nucleolus (**Fig.1c,6c in the revised manuscript**). The cavities in the LIN28A inclusions tended to encapsulate DFC(FBL) (**Fig.1b,1c in the revised manuscript**). Loss of LIN28A resulted in disrupted stratification of nucleoli, but the disordered nucleoli were still presented (**Fig.5e in the revised manuscript**). These results demonstrated LIN28A itself is not sufficient to form nucleolus, but its loss impairs the integrity of nucleolus. We can consider LIN28A as a marker of nucleolar integrity, it acts as

a solid shell or scaffold to help stabilizing the existing nucleolar layered structure. In Figure 2, we can see the nucleolar LIN28A condensate was sensitive to the rRNA inhibition, and it could be used as a marker of nucleolar integrity. It's highly possible that disrupting rRNA indirectly affected LIN28A through first disrupting the nucleolar integrity. Regarding nucleosome, as in question #3, we think that LIN28A mainly plays a role in the establishment of DFC and GC in the nucleolus, but not in FC where the rDNA is located, nor a role in nucleosome. Thus this question is out of scope of our present study, we can explore it in future studies.

Reviewer #2 (Remarks to the Author):

I have no further comments for the paper.

RESPONSE: We thank the reviewer for his positive comment on our current work.

- 1 Ricci, M. A., Manzo, C., García-Parajo, M. F., Lakadamyali, M. & Cosma, M. P. Chromatin fibers are formed by heterogeneous groups of nucleosomes in vivo. *Cell* **160**, 1145-1158, doi:10.1016/j.cell.2015.01.054 (2015).
- 2 Hou, Z., Nightingale, F., Zhu, Y., MacGregor-Chatwin, C. & Zhang, P. Structure of native chromatin fibres revealed by Cryo-ET in situ. *Nature Communications* **14**, 6324, doi:10.1038/s41467-023-42072-1 (2023).
- 3 Sun, Z. *et al.* LIN28 coordinately promotes nucleolar/ribosomal functions and represses the 2C-like transcriptional program in pluripotent stem cells. *Protein Cell* **13**, 490-512, doi:10.1007/s13238-021-00864-5 (2022).
- 4 Lopez, N. *et al.* Deconstructing virus condensation. *PLOS Pathogens* **17**, e1009926, doi:10.1371/journal.ppat.1009926 (2021).